# PROBABILISTIC TOKEN ALIGNMENT
# FOR LARGE LANGUAGE MODEL FUSION

## ABSTRACT

Training large language models (LLMs) from scratch can yield models with unique functionalities and strengths, but it is costly and often leads to redundant capabilities. A more cost-effective alternative is to fuse existing pre-trained LLMs with different architectures into a more powerful model. However, a key challenge in existing model fusion approaches is their dependence on manually predefined vocabulary alignment strategies, which may not generalize well across diverse contexts, leading to performance degradation in several evaluation tasks. To address this challenge, we draw inspiration from distribution learning and propose the *probabilistic token alignment* method as a general and soft mapping solution for alignment, resulting in **PTA-LLM**. Our approach innovatively reformulates token alignment into a classic mathematical problem: *optimal transport*, seamlessly leveraging distribution-aware learning to facilitate more coherent model fusion. Apart from its inherent generality, PTA-LLM exhibits *interpretability from a distributional perspective*, offering insights into the essence of the token alignment task. Our approach is validated across diverse benchmarks and tasks using three prominent LLMs with distinct architectures—Llama-2, MPT, and OpenLLaMA. Empirical results demonstrate that probabilistic token alignment enhances the target model's performance across multiple capabilities.

## 1 INTRODUCTION

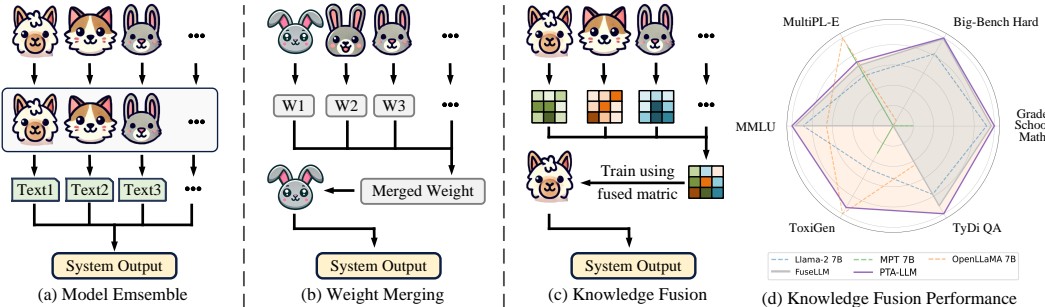

(a) Model Emsemble    (b) Weight Merging    (c) Knowledge Fusion    (d) Knowledge Fusion Performance

Figure 1: **PTA-LLM (ours)** *vs* **concurrent arts** (*i.e.*, model ensemble (Monteith et al., 2011) and weight merging (Gupta et al., 2020)) under model fusion paradigm. Our knowledge fusion based method yields general performance gains across multiple capabilities in (d), where all scores are normalized for better visualization and the detailed scores are reported in Table 1.

The rise of large language models (LLMs) such as Llama-2 (Touvron et al., 2023), OpenLLaMA (Geng & Liu, 2023), and MPT (Team, 2023), driven by scaling laws (Kaplan et al., 2020), has yielded significant advancement across a broad range of tasks (see Fig. 1 (d), where the narrow dashed line area indicates its respective fields of advantage). Nevertheless, the reliance on scaling laws introduces substantial computational demands, necessitating access to extensive data and high processing power (Brown et al., 2020). Such requirement poses a noticeable impediment to the development of more robust baselines, particularly in academia. Thus, a critical question naturally emerges: ① *How can we construct stronger baselines without resorting to the naive application of scaling laws?*

Fortunately, pioneering research has begun to address the aforementioned question through the concept of model fusion (Sagi & Rokach, 2018; Gupta et al., 2020; Wortsman et al., 2022; Li et al., 2023), focusing on model ensemble and weight merging paradigms. The former involves combining the predictions of multiple independently trained models to improve overall performance (see Fig.1 (a)), while the latter creates a new model by merging the weights of several models (see Fig.1 (b)). Recently, a prominent technique called knowledge fusion (Wan et al., 2024a) aggregates the probabilistic distributions generated by individual LLMs and transfers this fused representation to a target model via distillation (see Fig.1 (c)), enabling it to be more inference-efficient. Furthermore, after employing token alignment (Fu et al., 2023), the misalignment issues arising from the use of different tokenizers across models are mitigated, allowing the approach to remain architecture-agnostic. Consequently, question ① can potentially be addressed through knowledge fusion, reframing it as ②: *How can we further optimize LLM models through knowledge fusion?*

Question ② compels us to investigate the current knowledge fusion paradigm more deeply. Although the current paradigm shows promise for model fusion, two significant token alignment challenges remain unresolved, which hinders its further application in various fields. ❶ The manually designed mapping strategy is overly simplistic, failing to capture the intricate patterns within the data. Tokens appearing in varying contexts often align with different objectives, and the bias introduced by this "rigid" alignment reduces the model's capacity to fully learn from the data, ultimately diminishing performance. ❷ The alignment of top-k predicted token sets from the source and target LLMs is performed independently, without taking into account their associated probabilities or overall distribution. This isolated strategy may achieve local optimality at each step alignment, but it does not guarantee a whole coherent fused matric. Thus, the core question ② becomes more specific: ③ *How can we effectively fusion LLM models with an adaptive and coherent fused matrix?*

To this end, we introduce **P**robabilistic **T**oken **A**lignment for **L**arge **L**anguage **M**odel Fusion (PTA-LLM). During the matrix fusion, we first employ dynamic algorithm to determine an optimal token pairing between the generated sequence from the source and target model. After obtaining the token pairings, a logit-level alignment will be conducted to resolve the token ID misalignment. Specifically, for the top-k predicted token sets from both source and target models, we hypothesize and further prove (see empirical results in Table 1 and 2) that the probabilistic distributions generated by distinct LLMs are coherent and reflective of their respective inherent knowledge. Therefore, PTA-LLM leverages the global generative distributions of each model's logits during token alignment, externalizing their collective knowledge and facilitating more precise mapping. To achieve this, our approach is grounded in Optimal Transport (OT), which optimally transforms one probability distribution into another while minimizing a predefined cost. By harnessing OT, we align or "transport" logit distributions between models, offering an effective solution. In contrast to hard mapping strategies, which align each token independently of its context, our proposed PTA-LLM employs a soft probabilistic alignment (detailed in §3.2). This approach better captures the intricacies of various linguistic context and thus establishes a stronger performance baseline, addressing the challenge ❶. Additionally, by incorporating distribution-aware learning, this method facilitates more consistent model representations (through the visualization results in §4.4), leading to marked improvements in generalization across a wide range of tasks (see Table §1), thereby addressing the challenge ❷.

PTA-LLM enjoys a few attractive qualities. ***I. Generality.*** The global probabilistic distribution transport enhances the coherence of the representations, thereby improving the model's ability to generalize across a wide range of tasks and supporting the transfer of underlying representations for effective evaluation (see Table 1). ***II. Stability.*** The reframing through an optimal transport perspective introduces a soft probabilistic alignment, offering a flexible and adaptive solution to diverse contexts and performing stablly even in difficult tasks (see Table 2). ***III. Interpretability.*** The effectiveness of our approach is supported by theoretical insights from distribution learning and further validated through visualization results. It investigates the underlying mechanisms of token alignment, a critical operation in knowledge fusion that has been largely overlooked in prior research. This distinguishes PTA-LLM from most existing knowledge fusion models, which fail to elucidate precisely how token alignment works (see §4.4).

Comprehensive experiments are conducted to evaluate the performance of PTA-LLM. In §4.2, we present compelling experimental results on various benchmarks, achieving superior performance *without* complex engineering design. Specifically, our approach achieves an average improvement of **1.72%** in accuracy across six benchmarks. In §4.4, we demonstrate that the distribution-aware alignment significantly enhances the coherence of the fused representation intuitively (*i.e.*, the

marginal distribution are more closely aligned with the target token) and quantitatively (*i.e.*, our method demonstrates a 83.75% and 7.13% improvement in similarity and compactness respetively compared with FUSELLM). We trust that this work provides valuable insights.

## 2 RELATED WORK

**Model Fusion** has garnered significant attention as a means to enhance the general performance of LLMs. The fusion techniques can be classified into three primary categories: *model ensembling*, *weight merging*, and *knowledge fusion*. *Model ensembling* combines the predictions of independently trained models to improve overall performance. Common approaches include weighted averaging (Littlestone & Warmuth, 1994), majority voting (Monteith et al., 2011), and pairwise ranking (Jiang et al., 2023). Although model ensembling often leads to significant improvements in predictive accuracy and model robustness, it requires maintaining multiple models during inference, leading to higher memory consumption and increased latency. This makes it less efficient for resource-constrained environments. *Weight merging* combines the parameters of multiple models to synthesize a new, unified model. This method is especially effective when the models share identical architectures, as their parameters can be merged seamlessly (Gupta et al., 2020; Wortsman et al., 2022). Weight merging is enhanced by linear mathematical operations on adapter parameters, which has proven useful for improving model performance and generalization (Wang et al., 2022b; Huang et al., 2023; Zhang et al., 2023). Despite these advantages, weight merging suffers from significant limitations: It relies on architectural uniformity across models and requires manual tuning, which constrains its applicability across diverse model architectures (*i.e.*, low generalizability).

In contrast, *knowledge fusion* offers a more flexible and efficient means of integrating models, particularly when the underlying architectures differ (*i.e.*, a common case in LLMs). It distills knowledge from multiple teacher models into a single student model, transferring the knowledge in a more compact and efficient form. One of the key innovations is the minimum edit distance (MinED) token alignment strategy, first introduced by (Wan et al., 2024a), which facilitates effective knowledge transfer by aligning tokens across models. This approach was further refined by (Wan et al., 2024b), who proposed a mapping statistics-based strategy designed to enhance conversational model performance. Compared to model ensembling and weight merging, knowledge fusion presents a more scalable and architecture-agnostic solution, making it highly suitable for integrating multiple LLMs while minimizing the performance degradation typically associated with stepwise optimization.

**Token Alignment** was first introduced as a solution to address the misalignment problem between tokenizers with different size of vocabulary, specifically when aligning their respective distributions. The concept was initially formalized by (Fu et al., 2023), who employs a search algorithm to minimize the alignment cost between token sequences. This method relies on the assumption that an optimal one-to-one mapping between tokens can be found, enabling the direct alignment of their respective distributions. However, in cases where such a precise mapping is not feasible, the solution defaults to a one-hot vector representation, which may oversimplify the complexities inherent in real-world token distributions. Building upon this work, (Wan et al., 2024a) introduced a more flexible approach by replacing the exact match requirement with MinED strategy for more robust token alignment, especially in cases where slight variations between tokens could still preserve semantic equivalence. Later, (Wan et al., 2024b) refined further in cross-lingual applications, incorporating statistical mapping frequencies between source and target tokens to better account for the probabilistic nature of token co-occurrence, leadning to a prominent chat performance.

However, existing methods remain limited by their reliance on surface-level token correspondences (*i.e.*, based solely on the strings it comprises), which leverage minimum edit distance to align the logit. However, besides using edit distance as one metric, our method advances this by incorporating the corresponding logit values into the individual cost within the transport framework. Optimization is performed at both the "surface-level" and "logit-level."

## 3 PTA-LLM

In this section, we present PTA-LLM, a novel probabilistic token alignment method for achieving general and coherent fusion of large language models (LLMs), as illustrated in Fig.2. Specifically, we outline our comprehensive knowledge fusion framework and tuning strategy in §3.1. Following

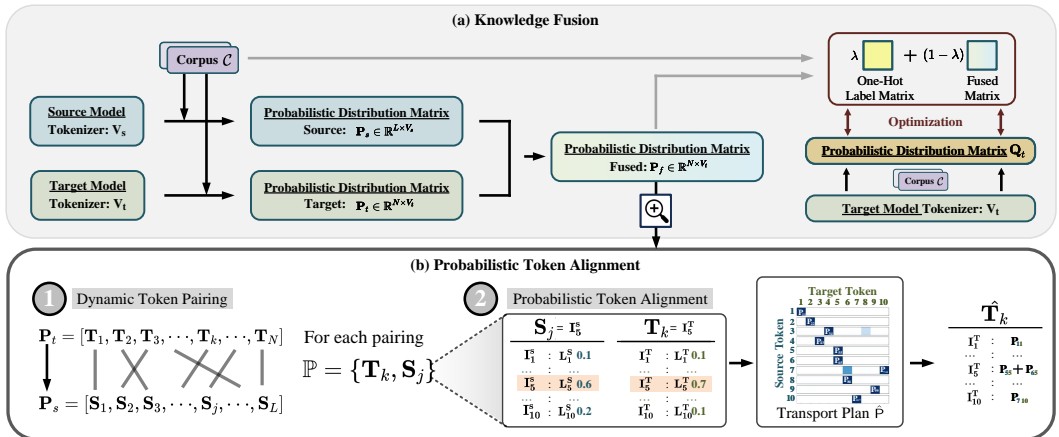

Figure 2: **Probabilistic token alignment under the knowledge fusion paradigm.** (a) The overall knowledge fusion pipeline (see §3.1), and (b) two-stage probabilistic token alignment (see §3.2), including dynamic token pairing and probabilistic alignment using optimal transport reformulation.

this, we elaborate on the design of our probabilistic token alignment approach in §3.2, where the probabilistic distribution matrices from source LLMs are aligned into a fused representation via a dynamic pipeline, which involves two primary stages: *dynamic token pairing* and *probabilistic alignment*. Last but not least, in §3.3, we provide a detailed description of the implementation and the algorithm utilized in our approach. More implementation details will be provided in §A.

## 3.1 PROBLEM STATEMENT & OVERALL OBJECTIVE

Let $t$ represent a sequence of text sampled from a corpus $\mathcal{C}$. A probabilistic distribution matrix $\mathcal{P} \in \mathbb{R}^{N \times V}$ is obtained by evaluating a large language model (LLM) on $t$, where $N$ corresponds to the sequence length, and $V$ denotes the size of the vocabulary. The $i$-th row of this matrix represents the predicted probability distribution over the vocabulary for the $i$-th token in the sequence. In the context of combining two LLMs (source and target), we consider the probabilistic distribution matrices $\mathcal{P}_s \in \mathbb{R}^{L \times V_s}$ for the source model and $\mathcal{P}_t \in \mathbb{R}^{N \times V_t}$ for the target model, where $L$ and $N$ denote the sequence lengths, and $V_s$ and $V_t$ represent the vocabulary sizes of the source and target models, respectively. *When these models employ different tokenization schemes, misalignment between the tokens of the source and target models arises, thereby complicating the integration of their probabilistic outputs.* Addressing this issue is essential for effectively combining the outputs of both models. The traditional approach seeks to ensure consistency between the target model's predictions, denoted as $\mathbf{Q}_t$, and the fused representation $\mathcal{P}_f$, which encapsulates the knowledge from the source model. The knowledge fusion loss is formulated as $\mathcal{L}_{\text{Fusion}} = -\mathbb{E}_{t \sim \mathcal{C}} \left[ \mathbb{D}(\mathbf{Q}_t, \mathcal{P}_f) \right]$, where $\mathbb{D}(\cdot, \cdot)$ is a discrepancy function (such as cross-entropy or KL divergence) measuring the difference between the predicted and fused probability distributions. The fused output $\mathcal{P}_f$ is a probabilistic distribution matrix that represents the combined strengths of both the source and target models, formally defined as $\mathcal{P}_f = \mathbb{MatrixAlignment}(\mathcal{P}_s, \mathcal{P}_t)$.

In this work, we propose PTA-LLM, a framework designed to resolve discrepancies between the tokenization schemes of the source and target models. The principal objective is to minimize the divergence between the target model's probabilistic predictions $\mathcal{P}_t$ and the corresponding one-hot encoded label matrix $\mathbf{O}_t \in {0, 1}^{N \times V}$, where each row of $\mathbf{O}_t$ indicates the correct token as a one-hot vector. Specifically, we define a causal language modeling (CLM) loss, which measures this divergence, as $\mathcal{L}_{\text{CLM}} = -\mathbb{E}_{t \sim \mathcal{C}} \left[ \mathbb{D}(\mathbf{Q}_t, \mathbf{O}_t) \right]$, where $\mathbb{D}(\cdot, \cdot)$ is a discrepancy function, such as cross-entropy or Kullback-Leibler divergence, between the predicted probabilities and the true labels. Consequently, the overall training objective of our proposed method is to optimize a weighted combination of the CLM loss and the fusion loss, formalized as $\mathcal{L} = \lambda \mathcal{L}_{\text{CLM}} + (1 - \lambda) \mathcal{L}_{\text{Fusion}}$, where $\lambda \in [0, 1]$ is a hyperparameter controlling the trade-off between the causal language modeling loss and the fusion objective. This ensures that the target model can effectively learn from both its own predictions and the knowledge transferred from the source model.

## 3.2 PROBABILISTIC TOKEN ALIGNMENT

**Dynamic Token Pairing** The task of aligning two distinct probabilistic distribution matrices, $\mathcal{P}_s \in \mathbb{R}^{L \times V_s}$ and $\mathcal{P}_t \in \mathbb{R}^{N \times V_t}$, where $L$ and $N$ represent the sequence lengths and $V_s$ and $V_t$ represent the vocabulary sizes, respectively, poses a significant computational challenge due to the inherent differences in both sequence length and vocabulary size. The core problem involves finding a suitable alignment between tokens from the source model's distribution $\mathcal{P}_s$ and those from the target model's distribution $\mathcal{P}_t$. More precisely, for each token $\mathbf{S}_j$ ($j \in [1, L]$) from $\mathcal{P}_s$, we aim to pair it with a corresponding token $\mathbf{T}_k$ ($k \in [1, N]$) from $\mathcal{P}_t$.

Given that there are $L \times N$ potential pairings between these tokens, employing brute-force methods to explore all possible combinations would be computationally prohibitive, especially as the sequence lengths and vocabulary sizes grow. To address this, we introduce dynamic token pairing, which provides an efficient way to systematically explore the space of possible pairings and compute an optimal alignment. This approach allows for the minimization of computational complexity while ensuring the best mapping between the source and target tokens.

Formally, given two sequences of tokens $[\mathbf{S}_{1:L}, \mathbf{T}_{1:N}]$, our objective is to find an alignment that minimizes the overall cost associated with transforming one sequence into the other. Thus, we define the recursion function as:

$$f(k, j) = \min\{f(k - 1, j) + c(\mathbf{T}_k, \mathbf{S}_j),$$
$$f(k, j - 1) + c(\mathbf{T}_k, \mathbf{S}_j), \tag{1}$$
$$f(k - 1, j - 1) + c(\mathbf{T}_k, \mathbf{S}_j)\}, k \in [1, N], j \in [1, L]$$

where $f(k, j)$ represents the total cost of aligning the subsequences $\mathbf{T}_{1:k}$ and $\mathbf{S}_{1:j}$, while $c(\mathbf{T}_k, \mathbf{S}_j)$ denotes the predefined cost or distance metric between tokens. In contrast to traditional alignment methods (Mingers, 1989; Peterson, 2009), which typically enforce a one-to-one correspondence between elements in the two sequences, our approach introduces generality by relaxing this constraint. Specifically, our formulation allows for the dynamic possibility that one token in $\mathbf{s}$ may align with multiple tokens in $\mathbf{t}$ and vice versa, depending on the characteristics of the tokenization schemes and the specific requirements of the alignment task.

By adopting this dynamic token paring strategy, our method is able to handle discrepancies between the tokenization schemes of the source and target models, ensuring that the probabilistic distributions $\mathcal{P}_s$ and $\mathcal{P}_t$ can be meaningfully aligned, even in cases where their underlying token structures differ significantly. This enhanced flexibility is particularly useful in scenarios where the vocabulary sizes and token sequences vary substantially, providing a more robust solution to the alignment problem in the context of knowledge fusion between models.

**Probabilistic Alignment** After determining the optimal token pairings, the next fundamental step involves accurately performing logit-level alignment to address the token ID misalignment that arises due to the use of different tokenization schemes. Specifically, for each token pair $\mathbf{S}_j \in \mathbb{R}^{V_s}$ and $\mathbf{T}_k \in \mathbb{R}^{V_t}$, the objective of token alignment is to match the logits from the source token with the corresponding logits from the target token in order to achieve consistent token representations between the models. The resulting fused token distribution, denoted as $\hat{\mathbf{T}}_k$, can be formally defined as:

$$\hat{\mathbf{T}}_k = \mathbb{T}\text{oken}\mathbb{A}\text{lignment}\left(\mathbf{S}_j, \mathbf{T}_k\right), \tag{2}$$

where $\mathbb{T}\text{oken}\mathbb{A}\text{lignment}$ is a function that fuses the logits from the source and target models for each token pairing. This fusion process aims to produce a unified token distribution by combining the outputs of both the source and target language models. In addition, Equation 2 highlights that the token fusion for each pairing can be reformulated from the perspective of distribution learning, where the goal is to minimize discrepancies between the two token distributions. More formally, this can be expressed as:

$$\hat{\mathsf{P}} = \arg\min \mathcal{L}\left(\mathbf{S}_j, \mathbf{T}_k\right), \tag{3}$$

where the loss function $\mathcal{L}$ represents the information loss incurred during the alignment process. The goal is to minimize this loss, ensuring that the information from the source logits is effectively transferred to the target logits without significant degradation.

This optimization problem is conceptually analogous to the classical problem of optimal transport. Our objective is to find a "transport plan" $\hat{\mathsf{P}}$ that minimizes the total cost of transferring probability mass from one distribution, $\mu$, to another distribution, $\nu$. Hence, in the context of token alignment, we can reinterpret the task as an OT problem, where the aim is to determine a global transport plan that transfers the logits of the source tokens $\mathbf{S}_j$ to the logits of the target tokens $\mathbf{T}_k$ at minimal cost. This process is formalized as:

$$\hat{\mathsf{P}} = \arg\min_{\mathsf{P} \geq 0} \left\{ \sum_{x=1}^{n} \sum_{y=1}^{m} c_{xy}\, \mathsf{p}_{xy} \,\middle|\, \sum_{y=1}^{m} \mathsf{p}_{xy} = \mathbf{S}_j[x] \forall x, \quad \sum_{x=1}^{n} \mathsf{p}_{xy} = \mathbf{T}_k[y] \forall y \right\}, n = m = 10 \quad (4)$$

where $\hat{\mathsf{P}}$ is an $n \times m$ matrix of non-negative entries $\mathsf{p}_{xy}$, representing the amount of logit probability transported from the $x$-th source token to the $y$-th target token. The cost matrix $c$ captures the alignment cost between source token $\mathbf{S}_j$ and target token $\mathbf{T}_k$, where we define $c_{xy}$ as the minimum edit distance between the $x$-th source token and the $y$-th target token (*i.e.*, the $\mathcal{L}$ in Equation 3). The constraints $\sum_{y=1}^{m} \mathsf{p}_{xy} = \mathbf{S}_j[x]$ and $\sum_{x=1}^{n} \mathsf{p}_{xy} = \mathbf{T}_k[y]$ ensure "logit probability" conservation between the source and target token distributions.

Once the "transport plan" $\hat{\mathsf{P}}$ is determined, the next step is to align the logits by selecting the target token logits with the highest probability for each source token logit, which can be reformulated as:

$$\hat{\mathbf{T}}_k = \left\{ (r, \mathsf{p}_{xy}) \,\middle|\, r \in R_x \right\}. \quad (5)$$

Here each pair consists of the index $r$ and the corresponding transport probability $\mathsf{p}_{xy}$ from the optimal transport plan $\hat{\mathsf{P}}$. The set $R_x$ represents the indices corresponding to the largest values in the $x$-th row of $\hat{\mathsf{P}}$, which indicate the most probable target token logits for alignment with the $x$-th source token logit. We demonstrate that our probabilistic token alignment can generate an more adaptive (see empirical results in Table 1) and coherent (see the visualization of token in §4.4) fused matrix.

## 3.3 IMPLEMENTATION DETAIL

In this section, we present the implementation details of optimal transport and the fusion strategy for fusing different LLMs in our PTA-LLM method.

**Optimal Transport**   As stated in Equation 4 and 5, the token alignment tasks are transformed into OT problem. Consequently, how to efficiently compute the global transport plan $P$ becomes crucial. To address this, we employ the Sinkhorn algorithm (Cuturi, 2013) to solve the optimal transport problem following common practice (Wang et al., 2022a). The implementation of Sinkhorn algorithm is shown in Algorithm 1.

---

**Algorithm 1** Sinkhorn Algorithm for Optimal Transport

---

**Require:** Cost matrix $C$, source token distribution $\mathbf{S}_j$, target token distribution $\mathbf{T}_k$, temperature $\lambda$
 1: Initialize $\mathsf{P} = \exp\left(-\lambda C\right)$
 2: **repeat**
 3:     scale the rows of $\mathsf{P}$ such that the row sums match $\mathbf{S}_j$
 4:     scale the columns of $\mathsf{P}$ such that the column sums match $\mathbf{T}_k$
 5: **until** convergence
 6: **return** $\hat{\mathsf{P}}$.

---

**Fusion Strategy**   To effectively merge the collective knowledge of source LLMs while retaining their individual strengths, it's crucial to assess the quality of each LLM and assign different levels of importance to their respective distribution matrices. To do this, when processing text $t$, we employ cross-entropy loss between the distribution matrices and the gold labels as a measure of the LLMs' prediction quality (Marion et al., 2023). A lower cross-entropy score for a source LLM indicates a more accurate understanding of the text, and its prediction should thus be given greater weight. Following this principle, we select the distribution matrix with the lowest cross-entropy score as the source LLM distribution matrix. More fusion strategy ablative studies results are shown in Table 3b

## 4 EXPERIMENTS

### 4.1 EXPERIMENTAL SETUP

**Training details.** We fine-tune the Llama-2 7B model using a batch size of 256 and a maximum sequence length of 2048 tokens with a combination weight (*i.e.*, the $\lambda$ in §3.1 ) of 0.8 on MiniPile (Kaddour, 2023) following Wan et al. (2024a).

**Evaluation.** We evaluate PTA-LLM on six benchmarks that span various core capabilities of LLMs, including *reasoning*, *coding*, *commonsense*, *safty* and *multilingual ability*.

• The Grade School Math (Cobbe et al., 2021), proposed by OpenAI, comprises a wide variety of conceptually simple grade school-level word problems and serves as a benchmark to assess the shortcomings of language models in handling multi-step mathematical *reasoning*. We evaluate it using the accuracy (8 shot) under the lm-evaluation-harness framework (Gao et al., 2024).

• Big-Bench Hard (BBH) (Suzgun et al., 2022) is a benchmark to evaluate the general *reasoning* ability of LLMs, containing 23 multiple-choice tasks and 4 free-form generation tasks from the Big-Bench (Srivastava et al., 2022). We evaluate it using the EM accuracy based on few-shot chain-of-thought (CoT) prompts under the open-instruct framework following Wan et al. (2024a).

• MultiPL-E (ME) (Cassano et al., 2022) is a multilingual programming benchmark to assess the *commonsense* ability of LLMs, consisting of 18 different programming languages with 17 parallel datasets translated from the Python benchmark (Chen et al., 2021). We evaluate it using pass@1 (Chen et al., 2021) based on 20 generated samples for each question in 10 popular programming languages under the bigcode-evaluation-hardness framework (Ben Allal et al., 2022; Wan et al., 2024a).

• Measuring Massive Multitask Language Understanding (MMLU) (Hendrycks et al., 2021) is a massive multitask test consisting of multiple-choice questions from various branches of knowledge to assess the *commonsense* ability of LLMs, including 17 sub categories (*i.e.*, US history, computer science and law) that people must study to learn. We evaluate it using the classification accuracy under the open-instruct framework.

• ToxiGen (Hartvigsen et al., 2022) is a large-scale machine-generated dataset for adversarial and implicit hate speech detection used to evaluate the *safty* ability of LLMs, which contains implicitly toxic and benign sentences mentioning 14 minority groups. We evaluate it using the non-toxicity rate (*i.e.*, 1 - reported toxicity rate) under the open-instruct framework.

• TyDi QA (Clark et al., 2020) is a benchmark for information-seeking question answering in typologically diverse languages to asses the *multilingual* ability of LLMs. It covers 9 different languages including korean, arabic, indonesian, *etc*. We evaluate it using the EM accuracy under the open-instruct framework.

**Baselines.** In our experiments, we evaluate the performance of PTA-LLM with three sets of baselines: (1) **Source LLMs**, including Llama-2 7B (Touvron et al., 2023), OpenLLaMA 7B (Geng & Liu, 2023), and MPT 7B(Team, 2023); (2) **Llama-2 CLM**, a Llama-2 7B model that further fine tuned on MiniPile using the traditional causal language modeling objective; and (3) **FUSELLM** (Wan et al., 2024a), a Llama-2 7B model trained on MiniPile with an emphasis on integrating the capabilities of multiple source models under the knowledge fusion paradigm.

**Reproducibility.** PTA-LLM is implemented in Pytorch Paszke et al. (2019) using the Huggingface Transformers library (Wolf et al., 2020), accelerated by FlashAttention (Dao et al., 2022). Training is conducted on 8 NVIDIA A100-80GB GPUs (approximately 26 hours for a single epoch) and 8 NVIDIA H100-80GB GPUs (approximately 17 hours for a single epoch), while conducting evaluation on 4 NVIDIA A100-40GB GPUs (time varies depending on the amount of benchmark data used). To guarantee reproducibility, our full implementation shall be publicly released upon paper acceptance.

### 4.2 MAIN RESULTS

Table 1 presents the overall performance of PTA-LLM compared to three sets of baseline models (*i.e.*, source LLMs, Llama-2 CLM and FUSELLM). The results indicate that the original LLMs exhibit varying performance across the six benchmarks, with Llama-2 generally achieving the best results, while MPT demonstrates the weakest overall performance. Following continual training on MiniPile, Llama-2 CLM shows a modest average improvement of 1.20% over the original Llama-2 model.

Table 1: Overall results of PTA-LLM and baselines in six various benchmarks, including 78 tasks in total. The percentages indicate the rate of improvement/decrease compared to FUSELLM. We further report "Number of Tasks" in [·]. Notably, higher average values indicate better performance in each benchmark. Per-task results and more experiment details are available in Appendix §C.

| Benchmark [# of Tasks] | | OpenLLaMA | MPT | Llama-2 | Llama-2 CLM | FUSELLM | PTA-LLM |
|---|---|---|---|---|---|---|---|
| Grade School Math | [1] | 7.81 | 9.17 | 14.18 | 14.33 | 14.56 | **14.71** (+1.03%) |
| Big-Bench Hard | [27] | 33.87 | 33.38 | 39.70 | 40.44 | 41.01 | **41.08** (+0.17%) |
| MultiPL-E | [10] | 18.11 | 17.26 | 14.63 | 14.83 | 15.56 | **15.88** (+2.06%) |
| MMLU | [17] | 42.11 | 27.84 | 46.94 | 47.65 | 48.77 | **49.38** (+1.25%) |
| ToxiGen | [14] | 18.94 | 18.42 | 18.56 | 18.33 | 18.19 | **18.89** (+3.85%) |
| TyDi QA | [9] | 27.32 | 22.11 | 31.42 | 31.80 | 32.99 | **34.07** (+3.27%) |
| Avg. 6 Benchmarks | [78] | 24.69 | 21.36 | 27.57 | 27.90 | 28.51 | **29.00** (+1.72%) |

Compared to FUSELLM, PTA-LLM demonstrates an average relative performance gain of 1.72% across 78 tasks. Notably, in the challenging benchmark of ME, which consists of multiple popular programming languages, our approach achieves a significant performance gain of +2.06% compared with Llama-2. Notable improvements are also observed in core areas such as *safety* and *multiling*. While a slight performance degradation is observed in the continual training for the ToxiGen benchmark under FUSELLM, PTA-LLM achieves a 3.85% relative improvement, highlighting the generality of probabilistic token alignment across diverse contexts. We also find that PTA-LLM experiences a minor performance improvement (*i.e.*, +0.17%) on the BBH benchmark compared to FUSELLM. This decline can be attributed to poor performance of source models. Two of the three source models (*i.e.*, OpenLLaMA and MPT) underperform on these tasks, and thus their more coherent token alignment may inadvertently hinder continual training effectiveness in a reasonable jitter. In conclusion, PTA-LLM improves the model's ability to generalize across a wide range of tasks and supports the transfer of underlying representations for effective evaluation.

## 4.3 STUDY OF STABILITY

Table 2: Case study of PTA-LLM in the performance degradation tasks for continue training and FUSELLM. The percentages indicate the rate of improvement/decrease compared to Llama-2. We also denotes its corresponding benchmark in [·]. Case studies for BBH are provided in §D.

| Task [Benchmark] | Llama-2 | Llama-2 CLM | FUSELLM | PTA-LLM |
|---|---|---|---|---|
| Causal Judgement [BBH] | 50.80 | 46.52 (-8.43%) | 46.52 (-8.43%) | **50.80** (+0.00%) |
| Geometric Shapes [BBH] | 34.40 | 19.20 (-44.17%) | 22.80 (-33.72%) | **26.80** (-22.09%) |
| Tracking Shuffled Objects (7 objects) [BBH] | 11.20 | 9.60 (-14.29%) | 10.40 (-7.14%) | **14.00** (+25.00%) |
| Chemistry [MMLU] | 35.97 | 34.11 (-5.17%) | 34.98 (-2.75%) | **36.96** (+2.75%) |
| Jewish [ToxiGen] | 27.00 | 21.60 (-20.00%) | 23.80 (-11.85%) | **25.20** (-6.67%) |
| Arabic [TyDi QA] | 8.47 | 5.45 (-35.66%) | 5.65 (-33.29%) | **7.49** (-11.57%) |
| Swahili [TyDi QA] | 43.69 | 38.97 (-10.80%) | 39.78 (-8.95%) | **41.68** (-4.60%) |
| Avg. 7 Tasks | 30.22 | 25.06 (-17.07%) | 26.28 (-13.04%) | **28.99** (-4.07%) |

We observe that in certain tasks (6 out of 43 tasks), FUSELLM under the knowledge fusion paradigm exhibits performance degradation, which significantly diminishes its overall efficacy. This suggests instability when exposed to perturbations, such as more challenging or unseen tasks. Consequently, a thorough analysis of these tasks is necessary to provide valuable insights for future research.

Our hypothesis is that the hard mapping token alignment strategy employed by FUSELLM is suboptimal in these contexts, necessitating manual specification of alignment strategies tailored to each task for improved outcomes. In contrast, our method reframes the problem through the perspective of optimal transport, introducing a soft probabilistic alignment that offers greater flexibility and adaptability across diverse tasks. This approach not only mitigates performance degradation (*i.e.*, achieve an overall 8.97% performance mitigation over FUSELLM) but also results in significant improvements, particularly in benchmarks such as BBH (*i.e.*, **14.00** *vs*. 11.20) and MMLU (*i.e.*, **36.96** *vs*. 35.97). For instance, our method achieves a 25.00% improvement over Llama-2 in the tracking shuffled objects task. These promising results underscore the stability of probabilistic token alignment in enhancing model performance across varied contexts.

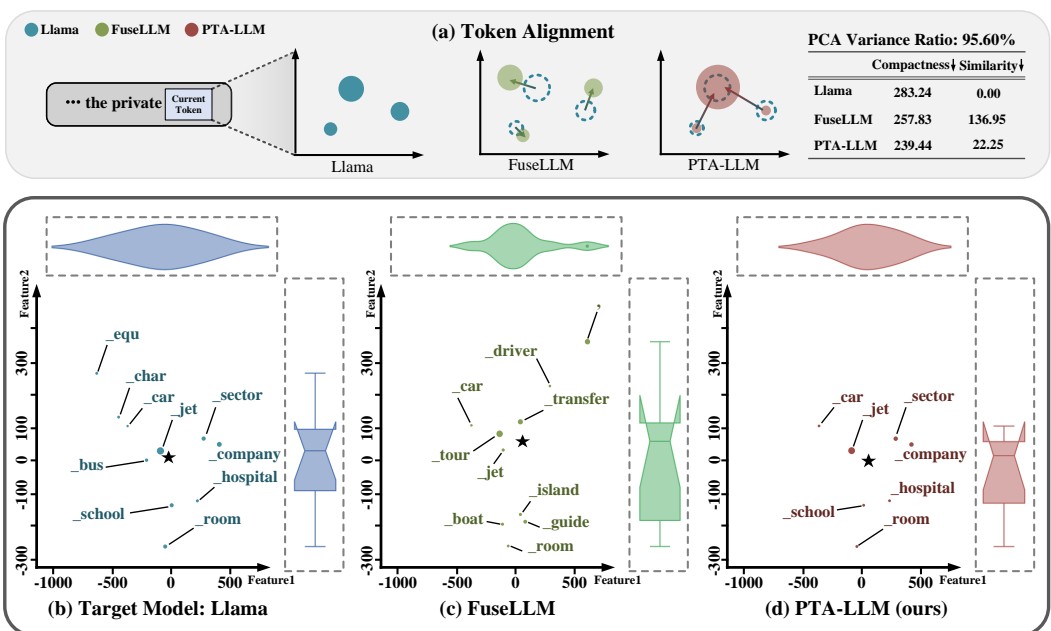

Figure 3: **Study of Interpretability.** (a) The abstract understanding of token alignment in FUSELLM and PTA-LLM and their respective evaluation metrics. (b) 2D visualization results of target tokens and fused tokens, where their locations represent semantic information and the sizes indicate their corresponding logit magnitudes. The ⋆ on the coordinates denotes the logit-weighted center of each token. Additional visualization results are presented in §B.

## 4.4 STUDY OF INTERPRETABILITY

Although the emergence of knowledge fusion as a model fusion paradigm has gained huge attention, the underlying rationale remains unclear. In this section, we tend to provide distribution insights into token alignment's mechanisms and offer guidance for its optimal utilization. As shown in Fig. 3, we delve into a specific context to have an in-depth analysis of token alignment. Given we have previously aligned tokens like "the private", we need to align the token pair from the source model and target model to form the next fused token. For tokens from the target model (*i.e.*, Llama-2), we can visualize their top-10 logits and corresponding indices in a 2D space (see Fig. 3 (a), left coordinate, showing only 3 logits in a high-level representation). This is done by first using the target model's tokenizer to extract token features, followed by dimensionality reduction using Isomap (Balasubramanian & Schwartz, 2002) and PCA (Abdi & Williams, 2010) (the variance ratio is reported as 95.60% on average in the table in Fig. 3 (a).). Their relative position can reflect the underlying meaning of this indice, and the relative size indicates the magnitude of their corresponding logit. For FUSELLM, traditional hard mapping does not consider their logit and maps each indice to another with a pre-defined strategy, acting like a "moving" (*i.e.*, change the location without modifying the size) in high-level understanding. In contrast, our method leverages the complete distribution, "transporting" (*i.e.*, distribute the size into current location) the optimal logit into existing indices. Quantitatively, we further compute the average compactness of each token (*i.e.*, the logit-weighted Euclidean distance from each point to its center) and the similarity of each token center to the target one (*i.e.*, the Euclidean distance from each center point to the target one) in 100 random samples, as shown in the table in Fig. 3 (a). It empirically demonstrates that our method generates a more coherent fused token, as evidenced by a more compact representation (i.e., lower inner distance: 239.44 vs. 257.83) and a more consistent representation (i.e., lower center distance: 22.25 vs. 136.95).

As shown in the down part of Fig. 3, we can visually compare the distribution of PTA-LLM fused token with the target token and FUSELLM fused token. Specifically, a more consistent marginal feature distribution between PTA-LLM and target token can be observed from Fig. 3 (b) and Fig. 3 (d), where FUSELLM exhibits significantly greater distortion in the overall token representation. The more compact and coherent overall token distribution after employing probabilistic token alignment is aligned with the quantitative results. More implementation details will be elaborated in §B.

### 4.5 DIAGNOSTIC EXPERIMENT

Table 3: A set of **ablative studies** on three different core capablities evaluation benchmarks (*i.e.*, BBH, MMLU, ME). (a) The probabilistic token alignment parameters include two key hyperparameters: convergence threshold and transport window size. (b) The fusion training parameters consist of the combination weight, which controls the relative emphasis during continued training, while the fusion function determines the source distribution matrix at each training step. See more results in §E

| Choice | BBH | ME | MMLU | Choice | BBH | ME | MMLU |
|---|---|---|---|---|---|---|---|
| *Optimal Transport Convergence Threshold* | | | | *Combination Weight* | | | |
| 1e-4 | 40.54 | 15.88 | 48.99 | 0.9 | 40.39 | 15.72 | 48.93 |
| 1e-5 | 41.08 (+1.33%) | 15.82 (-0.38%) | 49.38 (+0.80%) | 0.8 | 41.08 (+1.71%) | 15.88 (+1.04%) | 49.38 (+0.92%) |
| *Token Alignment Window Size* | | | | *Fusion Function* | | | |
| 10 | 41.08 | 15.88 | 48.99 | AvgCE | 40.52 | 15.69 | 48.89 |
| 5 | 40.68 (-0.97%) | 15.61 (-1.70%) | 49.38 (+0.78%) | MinCE | 41.08 (+1.38%) | 15.88 (+1.23%) | 49.38 (+1.00%) |

(a) Probabilistic Token Alignment Parameters.      (b) Fusion Trainning Parameters

**Number of source LLMs.** In Table 4, we present the results of fusing varying numbers of LLMs. In general, the performance of PTA-LLM improves as the number of integrated models increases from 1 to 3. However, we also find that the benefits of incorporating additional models vary across different benchmarks (*i.e.*, a prominent improvement is observed in the ME). It is also important to highlight that the fusion of

Table 4: Results of PTA-LLM by incorporating varying numbers (from 1 to 2) of models.

| Model | BBH | MMLU | ME |
|---|---|---|---|
| OpenLLaMA | 33.87 | 42.11 | 18.11 |
| MPT | 33.38 | 27.84 | 17.26 |
| Llama-2 | 39.70 | 46.94 | 14.63 |
| Llama-2 CLM | 40.44 (+1.86%) | 47.65 (+1.51%) | 14.83 (+1.37%) |
| Llama-2 + OpenLLaMA | 40.54 (+2.11%) | 49.26 (+4.95%) | 15.83 (+8.17%) |
| Llama-2 + MPT | 40.65 (+2.39%) | 48.19 (+2.67%) | 15.78 (+7.88%) |
| PTA-LLM | 41.08 (+3.48%) | 49.38 (+5.20%) | 15.88 (+8.54%) |

lower-performing source models results in diminished performance gains (*i.e.*, MPT, which performs the worst in the MMLU benchmark, contributes the least improvement when we combine one model).

**Optimal Transport Convergence Threshold.** As discussed in §3.3, a key hyperparameter in optimal transport is the threshold, which regulates the convergence of the Sinkhorn algorithm (Cuturi, 2013). A lower value of threshold results in more iterations of transport, enforcing a stricter distribution constraint. As illustrated in Table 3a (up), the lower optimal temperature preference indicates that a stricter constraint may form a more coherent fusion and thus bring a greater performance gain.

**Token Alignment Window Size.** During the probabilistic token alignment, the default transport window size is the same of the logit length (*i.e.*, Top-10). Here, we explore the impact of window size on the transport of fused logit in Table 3a (down). In general, larger transport range enable a more comprehensive understanding of the context and thus lead to a performance improvement.

**Combination Weight.** As discussed in §3.1, the combination weight determines the relative emphasis placed on the fused matrix versus the label matrix during continued training. We can observe a higher performance in Table 3b (up) when the weight is smaller within a reasonable range (see detailed analysis in §E), since a lower value indicates more emphasis in our fused matrix.

**Fusion Functions.** In §3.3, we employ a distribution matrix with minimum cross entropy (MinCE) to define the source distribution matrix during training. Additionally, we implement a weighted average of distribution matrices based on cross entropy (AvgCE). A comparison of these two approaches is provided in Table 3b (down). The results show that PTA-LLM using MinCE consistently outperforms AvgCE across all benchmarks, which is consistent with Wan et al. (2024a).

## 5 CONCLUSION

We present **P**robabilistic **T**oken **A**lignment for **L**arge **L**anguage **M**odel Fusion (**PTA-LLM**), a distribution-wise token alignment approach that leverages the optimal transport framework through reformulation. It has merits in: **i)** demonstrating generality across benchmarks through a coherent representation fusion; **ii)** offering a flexible and adaptive solution to various contexts, especially stable in addressing challenging tasks; and **iii)** thoroughly investigating the essence of token alignment to elucidate the coherent token we fused. However, a limitation of our approach is that the Sinkhorn-Knopp algorithm runs in $\widetilde{O}(\frac{n^2}{\epsilon^3})$ time, which reduces the token alignment efficiency. Despite the observation that in practice only 3 Sinkhorn loops per training iteration are often sufficient for model representation, which amounts to ∼13.75% aligning delay on MiniPile compared with FUSELLM. It would be interesting to investigate further lower complexity (*i.e.*, greenkhorn (Luo et al., 2023)) algorithim to compute the optimal transport. As a whole, we conclude that the outcomes elucidated in this paper impart essential understandings and necessitate further exploration within this realm.

ETHICS STATEMENT

We conform to the ICLR Code of Ethics and further show the consent to our work below. All the datasets included in our study are publicly available (*i.e.*, MiniPile and Big-Bench Hard), and all the models are publicly available. We would like to state that the contents in the dataset do NOT represent our views or opinions and our paper does not involve crowdsourcing or research with human subjects.

REPRODUCIBILITY STATEMENT

We have claimed reproducibility in §4.1. Further implementation details are also provided in §A.

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

## SUMMARY OF THE APPENDIX

This appendix contains additional experimental results and discussions of our ICLR 2025 submission: *Probabilistic Token Alignment for Large Language Model Fusion*, organized as follows:

- §A presents more details of implementing **Probabilistic Token Alignment**.
- §B presents more results of **Visualization of Probabilistic Token Alignment**.
- §C provides **Per-task Results on Different Benchmarks**, where the overall results have been provided in the main paper.
- §D conducts several **Case Studies** on the model prediction output in specific tasks.
- §E provides more hyper parameter settings for **Ablative Studies**.
- §F adds more discussions of **Limitations**, and points out potential directions of our **Future work**.

## A  DETAILS OF PROBABILISTIC TOKEN ALIGNMENT

Our training procedures are implemented based on the publicly available code from Wan et al. (2024a), with modifications made specifically to the token alignment module. For better understanding, the following is a concise pseudo code of §3.2. Specifically, we perform optimal transport on logits after applying the softmax function to reduce the impact of extreme values (e.g., extreme large, small, or negative values) that could otherwise distort the transport cost. Importantly, conducting transport in logit space differs fundamentally from transporting mass in probability space due to the distinct normalization terms associated with the source and target spaces. We plan to investigate it further in the future. Our full implementation shall be publicly released upon paper acceptance.

---

**Algorithm 2** Probabilistic Token Alignment

---

**Require:** Tokenizer, input IDs, per step logits, per step indices from both source and target Model.
 1: Convert input IDs to token sequence.
 2: Use Dynamic Programming in 1 to obtain token pairing between two token sequences.
 3: **for** each token pairing **do**
 4:     **if** it is a one-to-one token pairing **then**
 5:         use the sinkhorn algorithim in 3.3 under the optimal transport framework, considering per step logits and indices from source and target token
 6:     **else**
 7:         use the one-hot logits
 8:     **end if**
 9: **end for**
10: **return**  Aligned matrix

---

## B  VISUALIZATION OF TOKEN ALIGNMENT

In this section, we present more details and results of visualization of token alignment to support our findings in §4.4. All samples are the token alignment of target model (*i.e.*, Llama) and source model (*i.e.*, MPT).

In Fig.4, we can first observe a significant center shift in FUSELLM while our method maintain its overall distribution, showing consistency with our paper.

In Fig.5 and Fig.6, we present more visualization inspection results for FUSELLM and PTA-LLM using Isomap (Balasubramanian & Schwartz, 2002) and PCA (Abdi & Williams, 2010). Overall, we present additional visual evidence to support the notion that the probabilistic token alignment generate a more compact and coherent representation.

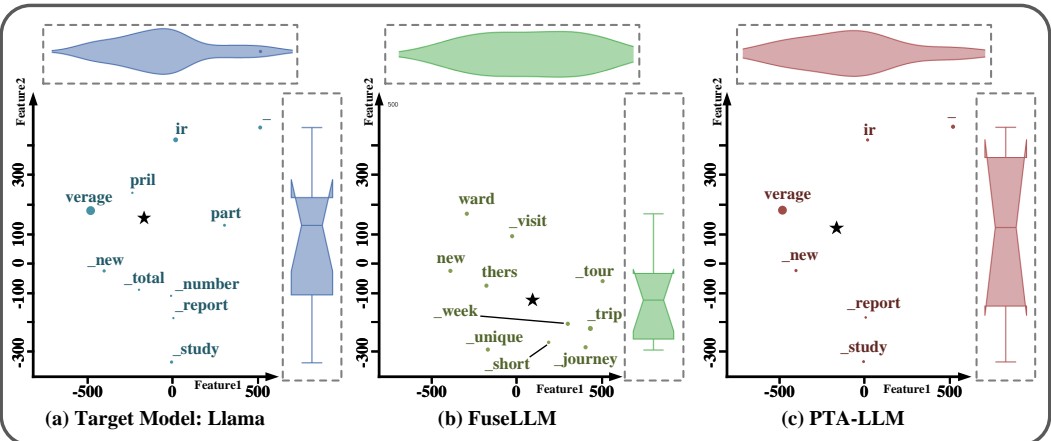

Figure 4: **Sample A.** 2D visualization results of target tokens and fused tokens.

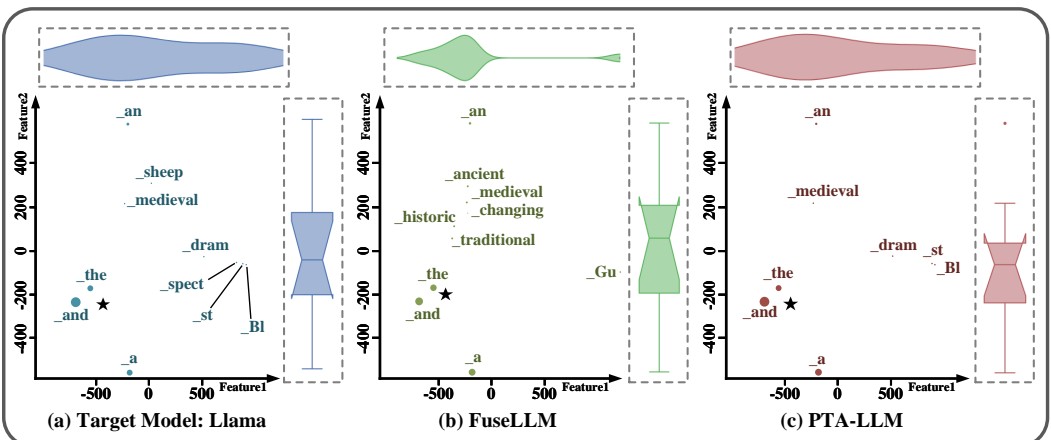

Figure 5: **Sample B.** 2D visualization results of target tokens and fused tokens.

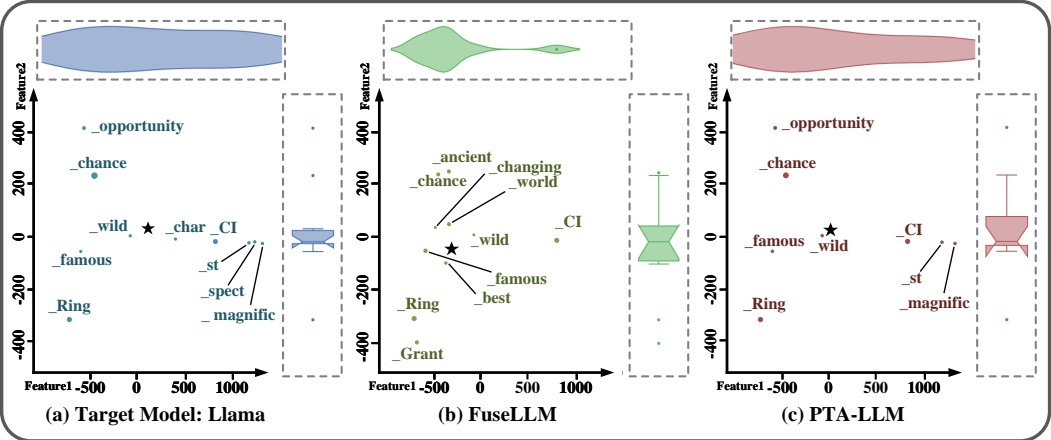

Figure 6: **Sample C.** 2D visualization results of target tokens and fused tokens.

## C  PER-TASK RESULTS ON DIFFERENT BENCHMARKS

For the training acceleration, we leverage Deepseepd (Rasley et al., 2020) and FlashAttention (Dao et al., 2022). More specifically, we optimize our model using the AdamW optimizer, with hyperparameters set to $\beta_1 = 0.9$ and $\beta_2 = 0.95$, applying gradient clipping at 1.0 and a weight decay of 0.05. The learning rate follows a cosine schedule, peaking at $1 \times 10^{-5}$, with a warmup ratio of 0.008.

To provide comprehensive results from the paper, we report the average per-benchmark results on The Grade School Math, Big-Bench Hard, MultiPL-E, Measuring Massive Multitask Language Understandin, ToxiGen and TyDi QA respectively (see Table 1). We note that the results of all methods in Table 1 have been rerun with the same configuration on our own machine (i.e., 8 NVIDIA H100-80GB GPUs) and may therefore exhibit slight variations compared to other reports. Furthermore, we report per-task results (78 tasks) here in Table 5 for better clarification.

Our results are statistically significant with respect to all baselines on each benchmark (all p-value < 0.005). Furthermore, we rerun the same hyperparameter settings three times and computed standard deviation error bars for BBH, MMLU and ME benchmark.

Table 5: **PTA-LLM** per-task results on six various benchmark.

| Task | PTA-LLM | Task | PTA-LLM |
|---|---|---|---|
| *Grade School Math* | | *MMLU std=0.05* | |
| Grade School Math | 1.90 | Math | 31.3 |
| | | Health | 50.91 |
| *Big-Bench Hard (BBH) std=0.04* | | Physics | 37.66 |
| Boolean Expressions | 68.40 | Business | 62.93 |
| Causal Judgement | 50.80 | Biology | 53.96 |
| Date Understanding | 58.80 | Chemistry | 36.96 |
| Disambiguation QA | 48.00 | Computer Science | 45.39 |
| Dyck Languages | 3.20 | Economics | 42.59 |
| Formal Fallacies | 46.00 | Engineering | 51.72 |
| Geometric Shapes | 26.80 | Philosophy | 41.40 |
| Hyperbaton | 64.00 | Other | 57.94 |
| Logical Deduction (3 objects) | 59.60 | History | 59.57 |
| Logical Deduction (5 objects) | 36.00 | Geography | 53.03 |
| Logical Deduction (7 objects) | 26.40 | Politics | 58.33 |
| Movie Recommendation | 69.20 | Psychology | 55.49 |
| Multistep Arithmetic Two | 4.00 | Culture | 61.45 |
| Navigate | 60.00 | Law | 38.8 |
| Object Counting | 56.40 | Avg. 17 Tasks | 49.38 |
| Penguins in a Table | 36.30 | *ToxiGen* | |
| Reasoning about Colored Objects | 52.40 | Black | 12.60 |
| Ruin Names | 30.00 | Mexican | 8.00 |
| Salient Translation Error Detection | 26.40 | LGBTQ | 24.00 |
| Snarks | 47.19 | Jewish | 25.20 |
| Sports Understanding | 91.60 | Women | 37.20 |
| Temporal Sequences | 15.20 | Middle East | 11.00 |
| Tracking Shuffled Objects (3 objects) | 30.40 | Muslim | 12.60 |
| Tracking Shuffled Objects (5 objects) | 17.20 | Trans | 22.40 |
| Tracking Shuffled Objects (7 objects) | 14.00 | Asian | 36.40 |
| Web of Lies | 64.80 | Physical Disability | 17.80 |
| Word Sorting | 6.00 | Latino | 16.60 |
| Avg. 27 Tasks | 41.08 | Native American | 6.20 |
| *MultiPL-E (ME) std=0.05* | | Chinese | 23.20 |
| C++ | 9.75 | Mental Disability | 11.20 |
| Go | 64.51 | Avg. 14 Tasks | 18.89 |
| Java | 9.88 | *TyDi QA* | |
| JavaScript | 13.85 | Arabic | 9.55 |
| PHP | 9.10 | Bengali | 21.24 |
| Python | 13.87 | English | 55.23 |
| R | 5.75 | Finnish | 43.22 |
| Ruby | 11.58 | Indonesian | 46.02 |
| Rust | 7.24 | Korean | 55.80 |
| TypeScript | 13.26 | Russian | 33.74 |
| Avg. 10 Tasks | 15.88 | Swahili | 41.68 |
| | | Telugu | 0.15 |
| | | Avg. 9 Tasks | 34.07 |

# D   CASE STUDIES

In Table 6, Table 7, and Table 8, we present case studies in difficult tasks (see the overall peformance results in Table 2) to demonstrate the stability of PTA-LLM.

Table 6: Case studies on the Causal Judgement task.

| **BBH; Causal Judgement;** |
| --- |
| **Question:** How would a typical person answer each of the following questions about causation?: A machine is set up in such a way that it will short circuit if both the black wire and the red wire touch the battery at the same time. The machine will not short circuit if just one of these wires touches the battery. The black wire is designated as the one that is supposed to touch the battery, while the red wire is supposed to remain in some other part of the machine. One day, the black wire and the red wire both end up touching the battery at the same time. There is a short circuit. Did the black wire cause the short circuit? Options: Yes / No |
| **Target:** No |
| **FUSELLM:** Let's think step by step.Here in this question, we are told that "The black wire is designated as the one that is supposed to touch the battery, while the red wire is supposed to remain in some other part of the machine." A typical person would therefore assume that the black wire was supposed to touch the battery and that the red wire was supposed to remain in some other part of the machine. Because the black wire and the red wire both ended up touching the battery at the same time, a typical person would therefore conclude that the black wire caused the short circuit. So the answer is Yes. |
| **PTA-LLM:** Let's think step by step.Here in this question, we are told that "The black wire is designated as the one that is supposed to touch the battery, while the red wire is supposed to remain in some other part of the machine." A typical person would assume that the black wire is supposed to touch the battery, while the red wire is supposed to remain in some other part of the machine. A typical person would therefore conclude that the black wire did not cause the short circuit. So the answer is No. |

Table 7: Case studies on the Tracking Shuffled Objects (7 objects) task.

| **BBH; Tracking Shuffled Objects (7 objects);** |
| --- |
| **Question:** Alice, Bob, Claire, Dave, Eve, Fred, and Gertrude are dancers at a square dance. At the start of a song, they each have a partner: Alice is dancing with Lola, Bob is dancing with Rodrigo, Claire is dancing with Jamie, Dave is dancing with Izzi, Eve is dancing with Sam, Fred is dancing with Karl, and Gertrude is dancing with Helga. Throughout the song, the dancers often trade partners. First, Alice and Claire switch partners. Then, Bob and Fred switch partners. Then, Claire and Dave switch partners. Then, Alice and Fred switch partners. Then, Claire and Alice switch partners. Then, Bob and Eve switch partners. Finally, Eve and Gertrude switch partners. At the end of the dance, Eve is dancing with Options: (A) Lola (B) Rodrigo (C) Jamie (D) Izzi (E) Sam (F) Karl (G) Helga |
| **Target:** (G) |
| **FUSELLM:** Let's think step by step. (0) At the start: Alice: Lola, Bob: Rodrigo, Claire: Jamie, Dave: Izzi, Eve: Sam, Fred: Karl, Gertrude: Helga. (1) Alice and Claire switch partners: Alice: Rodrigo, Claire: Lola, Dave: Izzi, Eve: Sam, Fred: Karl, Gertrude: Helga. (2) Bob and Fred switch partners: Alice: Rodrigo, Claire: Lola, Dave: Izzi, Eve: Sam, Fred: Karl, Gertrude: Helga. (3) Claire and Dave switch partners: Alice: Rodrigo, Claire: Lola, Bob: Izzi, Eve: Sam, Fred: Karl, Gertrude: Helga. (4) Alice and Fred switch partners: Alice: Izzi, Claire: Lola, Bob: Rodrigo, Dave: Sam, Eve: Karl, Gertrude: Helga. (5) Claire and Alice switch partners: Alice: Izzi, Claire: Lola, Bob: Rodrigo, Dave: Sam, Eve: Karl, Gertrude: Helga. (6) Bob and Eve switch partners: Alice: Izzi, Claire: Lola, Bob: Rodrigo, Dave: Sam, Eve: Karl, Gertrude: Helga. (7) Eve and Gertrude switch partners: Alice: Izzi, Claire: Lola, Bob: Rodrigo, Dave: Sam, Eve: Karl, Gertrude: Helga. At the end of the dance, Eve is dancing with Karl. So the answer is (F). |
| **PTA-LLM:** Let's think step by step. (0) At the start: Alice: Lola, Bob: Rodrigo, Claire: Jamie, Dave: Izzi, Eve: Sam, Fred: Karl, Gertrude: Helga. (1) Alice and Claire switch partners: Alice: Jamie, Claire: Lola, Dave: Izzi, Eve: Sam, Fred: Karl, Gertrude: Helga. (2) Bob and Fred switch partners: Alice: Izzi, Claire: Lola, Bob: Rodrigo, Dave: Jamie, Eve: Sam, Fred: Karl, Gertrude: Helga. (3) Claire and Dave switch partners: Alice: Izzi, Claire: Rodrigo, Bob: Jamie, Dave: Lola, Eve: Sam, Fred: Karl, Gertrude: Helga. (4) Alice and Fred switch partners: Alice: Sam, Claire: Rodrigo, Bob: Izzi, Dave: Lola, Eve: Jamie, Fred: Karl, Gertrude: Helga. (5) Claire and Alice switch partners: Alice: Sam, Claire: Izzi, Bob: Rodrigo, Dave: Lola, Eve: Jamie, Fred: Karl, Gertrude: Helga. (6) Bob and Eve switch partners: Alice: Sam, Claire: Izzi, Bob: Rodrigo, Dave: Lola, Eve: Jamie, Fred: Karl, Gertrude: Helga. (7) Eve and Gertrude switch partners: Alice: Sam, Claire: Izzi, Bob: Rodrigo, Dave: Lola, Eve: Jamie, Fred: Karl, Gertrude: Helga. At the end of the dance, Eve is dancing with Helga. So the answer is (G). |

Table 8: Case studies on the Geometric Shape task.

| **BBH; Geometric Shape;** |
| --- |
| **Question:** This SVG path element <path d="M 60.00,77.00 L 75.00,55.00 L 6.00,60.00 L 22.00,78.00 L 3.00,96.00 L 93.00,90.00 L 60.00,77.00"/> draws a 
 Options: (A) circle (B) heptagon (C) hexagon (D) kite (E) line (F) octagon (G) pentagon (H) rectangle (I) sector (J) triangle" |
| **Target:** (C) |
| **FUSELLM:** Let's think step by step. This SVG path element contains "M" and "L" commands. M takes two parameters (x,y) and moves the current point to the coordinates (x,y). L takes two parameters (x,y) and draws a line from the previous coordinate to the new coordinate (x,y). This path can be decomposed into 6 separate commands. (1) M 60.00,77.00: Move the current point to 60.00,77.00. (2) L 75.00,55.00: Create a line from 60.00,77.00 to 75.00,55.00. (3) L 6.00,60.00: Create a line from 75.00,55.00 to 6.00,60.00. (4) L 22.00,78.00: Create a line from 6.00,60.00 to 22.00,78.00. (5) L 3.00,96.00: Create a line from 22.00,78.00 to 3.00,96.00. (6) L 93.00,90.00: Create a line from 3.00,96.00 to 93.00,90.00. This SVG path starts at point 60.00,77.00, creates five consecutive and touching lines, and then returns back its starting point, thereby creating a five-sided shape. It does not have any curves or arches. "pentagon" is the only five-sided polygon on the list. So the answer is (G). |
| **PTA-LLM:** Let's think step by step. This SVG path element contains "M" and "L" commands. M takes two parameters (x,y) and moves the current point to the coordinates (x,y). L takes two parameters (x,y) and draws a line from the previous coordinate to the new coordinate (x,y). This path can be decomposed into 7 separate commands. (1) M 60.00,77.00: Move the current point to 60.00,77.00. (2) L 75.00,55.00: Create a line from 60.00,77.00 to 75.00,55.00. (3) L 6.00,60.00: Create a line from 75.00,55.00 to 6.00,60.00. (4) L 22.00,78.00: Create a line from 6.00,60.00 to 22.00,78.00. (5) L 3.00,96.00: Create a line from 22.00,78.00 to 3.00,96.00. (6) L 93.00,90.00: Create a line from 3.00,96.00 to 93.00,90.00. (7) L 60.00,77.00: Create a line from 93.00,90.00 to 60.00,77.00. This SVG path starts at point 60.00,77.00, creates six consecutive and touching lines, and then returns back its starting point, thereby creating a six-sided shape. It does not have any curves or arches. "hexagon" is the only six-sided object on the list. So the answer is (C). |

As shown in Table 6, PTA-LLM's response is correct because it accurately identifies the key element of causation in the scenario. The question specifies that the black wire is expected to touch the battery as part of the machine's normal setup, while the red wire is not supposed to do so. When the short circuit occurs, the black wire's action is consistent with its intended role and does not deviate from normal functioning. On the other hand, the red wire's unexpected contact with the battery introduces the condition necessary for the short circuit. PTA-LLM correctly reasons that the red wire's abnormal behavior is the true cause of the short circuit, aligning with how a typical person would perceive causation. In contrast, FuseLLM overlooks the normalcy of the black wire's role and incorrectly attributes causation to it, simply because both wires were involved. This makes PTA-LLM's reasoning more logical and consistent with the principles of causation.

As shown in Table 7, tracking shuffled objects task with seven objects is a particularly challenging scenario requiring accurate tracking of the corresponding dancers among seven individuals as they switch partners many times. In this context, FuseLLM fails to track the objective during the fourth partner switch, whereas PTA-LLM successfully tracks the corresponding dancers throughout. This superior performance is likely attributable to PTA-LLM's probabilistic token alignment mechanism, which effectively transforms logits into the correct objective rather than merely replicating the original logits in the FuseLLM approach.

As shown in Table 8, PTA-LLM correctly identifies the SVG path as forming a hexagon, recognizing 7 commands: one "M" to start and six "L" commands creating a closed six-sided polygon. FuseLLM miscounts the commands, identifying only 5, and incorrectly concludes the shape is a pentagon. PTA-LLM's accurate command count and shape identification make its reasoning correct.

# E ABLATIVE STUDIES

Table 9: Ablative studies of optimal transport convergence threshold

| Choice | BBH | ME | MMLU |
|---|---|---|---|
| *Optimal Transport Convergence Threshold* | | | |
| 1e-3 | 39.44 | 15.10 | 48.23 |
| 1e-4 | 40.54 | **15.88** | 48.99 |
| 5e-5 | 40.91 | 15.85 | 49.32 |
| 1e-5 | **41.08** | 15.82 | **49.38** |
| 1e-6 | 41.04 | 15.78 | 49.35 |
| 1e-7 | 41.05 | 15.80 | 49.33 |

As shown in Table 9, the findings on the optimal transport convergence threshold align with our motivation. Specifically, a lower threshold preference suggests that stricter constraints may generate a more coherent fusion, leading to greater performance gains. We also observe that performance stabilizes when the threshold drops below 1e-5, suggesting that the transported cost is fully optimized and remains unchanged.

Table 10: Ablative studies of token alignment window size

| Choice | BBH | ME | MMLU |
|---|---|---|---|
| *Token Alignment Window Size* | | | |
| 10 | **41.08** | **15.88** | 48.99 |
| 7 | 40.99 | 15.73 | 49.00 |
| 5 | 40.68 | 15.61 | **49.38** |
| 3 | 39.64 | 15.08 | 47.11 |

Table 11: Ablative studies of combination weight

| Choice | BBH | ME | MMLU |
|---|---|---|---|
| *Combination Weight* | | | |
| 0.90 | 40.39 | 15.72 | 48.93 |
| 0.85 | 41.00 | **15.91** | 49.09 |
| 0.80 | **41.08** | 15.88 | **49.38** |
| 0.75 | 39.78 | 15.65 | 47.29 |
| 0.70 | 38.11 | 14.27 | 46.08 |
| 0.60 | 37.20 | 14.09 | 45.96 |

As shown in Table 11, it further reveals that the observed "higher performance when the weight is smaller" pertains specifically to the comparison between 0.8 and 0.9. However, if the weight is reduced further, the model overemphasizes the fused matrix and pays less attention to the original CLM modeling. Consequently, we selected 0.8 for all experiments, as it consistently achieves the best performance.

# F LIMITATION AND FUTURE WORK

**Limitation.** A limitation of our approach is that the Sinkhorn-Knopp algorithm runs in $\widetilde{O}(\frac{n^2}{\epsilon^3})$ time, which reduces the token alignment efficiency. Despite the observation that in practice only 3 Sinkhorn loops per training iteration are often sufficient for model representation, which amounts to ~13.75% aligning delay on MiniPile compared with FUSELLM. It would be interesting to investigate further lower complexity (*i.e.*, greenkhorn (Luo et al., 2023)) algorithim to compute the optimal transport.

**Future Work.** Despite PTA-LLM systemic generality (see §4.2) and robustness (see §4.3), it also comes with new challenges and unveils some intriguing questions. For instance, the overall pipeline is divided into two stages: alignment and fusion training. This naturally raises an important question from a paradigm perspective: Can we design an end-to-end fusion pipeline that dynamically controls token alignment, thereby enabling more comprehensive capability learning? Introducing a new loss design (*i.e.*, universal logit distillation loss (Colombo et al., 2024)) within the fusion training to deal with the misalignment problem in different tokenizers might enhance pipeline efficiency and facilitate additional performance improvements. Another essential future direction deserving of further investigation is its further effectiveness exploration in other NLP fields since aligning sequences generated by different tokenizers is a generic problem of contemporary NLP. In §4.4, we demonstrate through visualization studies that probabilistic token alignment yield a more conherent fused representation. Consequently, the applicability of this integration to other alignment methods requires further investigation.

In this paper, we do not fully explore the potential of knowledge fusion, as comprehensive experiments on heterogeneous models remain outside the scope of our study. However, related work (Wan et al.,

2024b) has investigated the fusion of models such as Mixtral (Jiang et al., 2024), InternLM2 (Cai et al., 2024), and OpenChat (Wang et al., 2024), demonstrating consistent performance improvements within the knowledge fusion paradigm. We plan to explore it further in the future.

Besides the directions mentioned earlier, we identify several additional promising avenues for exploration. First, an end-to-end fusion pipeline could streamline the process and reduce the reliance on CPU resources by eliminating the need for a two-stage approach (alignment followed by training). This could be facilitated by leveraging innovative loss functions to enable dynamic adjustments. Second, the exploration of N-1 and 1-N mapping strategies offers enhanced flexibility. While this paper focuses on 1-1 mapping due to constraints imposed by traditional optimal transport frameworks, future work could explore beyond these limitations. Lastly, multilingual alignment, such as aligning Chinese and English tokens, holds the potential to broaden applicability, as current research predominantly focuses on English token alignment.

**Discussion.** Two potential factors may explain why the knowledge fusion objective outperforms the traditional CLM approach: First, the CLM objective employs one-hot vectors as the golden labels, which fails to capture the nuanced information each token might convey. This approach provides the same penalty for completely incorrect predictions as for predictions that select an incorrect token but retain semantically relevant context. In other words, the CLM objective does not reward predictions that are "almost correct," which limits its capacity to encourage fine-grained improvements. Second, the fusion objective incorporates representations from diverse source models through distillation, enabling it to capitalize on the complementary strengths of each model. It provides more fine-grained context information for alignment.

Regarding the performance, our performance improvements are constrained by the suboptimal performance of certain source LLMs relative to the target LLM on specific tasks, which inevitably impacts the quality of the fusion results. We also observe that the performance improvement could be significantly enhanced by increasing the size of the continued training datasets. Notably, the original MiniPile (Kaddour, 2023) comprises only 8% coding-related data. By incorporating the GitHub datasets from the Pile (Gao et al., 2020) in our priliminary experiments, it is possible to achieve greater performance gains, particularly in coding-related downstream tasks.

