# OpenReview forum: "Probabilistic Token Alignment for Large Language Model Fusion"
_ICLR.cc/2025/Conference — Submitted to ICLR 2025_

### Official Review · Reviewer_qifk · 2024-10-29

**Soundness:** 3
**Presentation:** 3
**Contribution:** 2
**Rating:** 5
**Confidence:** 4

**Summary:**

This paper introduces a novel approach PTA-LLM to fusing LLMs with different architectures by framing token alignment as an optimal transport problem. This distribution-aware method generalizes token alignment effectively, enhancing performance in zero-shot evaluation tasks and offering interpretability through a distributional lens.

**Strengths:**

This work addresses token alignment, a key challenge of knowledge fusion, by applying the classic mathematical optimization method of optimal transport to achieve a soft alignment. This approach also offers an interpretable analysis for token alignment.

**Weaknesses:**

1.This article focuses on token alignment for models with different structures but only tests three models, with OpenLLaMA and Llama2 having relatively high similarity. The experiment would benefit from including models with greater architectural diversity, such as Mistral and Qwen.

2.In Table 1 of the main experimental results, PTA-LLM shows limited improvement over FUSELLM, with an average absolute gain of less than 0.5 points. Additionally, the experiment employs a zero-shot evaluation approach, which is challenging for the base models and may not fully reflect their capabilities; for example, the GSM dataset typically uses an 8-shot or 4-shot setup. In the zero-shot setting, the scores of the three base models remain very low.

3.In Section 4.3 on STABILITY, the authors selected certain tasks to compare the performance of PTA-LLM and FUSELLM, demonstrating that soft alignment offers greater flexibility and adaptability than hard mapping. However, the article does not analyze the characteristics and shared features of these advantageous tasks or assess the applicability of soft alignment in these cases, making it difficult for the explanation to convincingly support its hypothesis.

4.In the ablation experiments of Section 4.5, the paper examines only two cases each for Optimal Transport Temperature, Combination Weight, and Token Alignment Window Size, which does not adequately reveal the impact trends of each factor.

5.There are some typos in the paper：

Line 223: "Pt" -> "P_t"

Line 274: "Tt" -> "T_t"

Line 417:  "-17/07%" -> "-17.07%"

Line 430: "26.96" -> "36.96"

**Questions:**

See "Weaknesses".

---

> ### Author Response · Authors · 2024-11-21
>
> Dear reviewer qifk,
>
> We sincerely appreciate your time and effort in reviewing our paper and providing valuable comments. We provide explanations to your questions point-by-point in the following.
>
> **Q1: Regarding the potential application to other source models.**
>
> Thank you for the excellent suggestion. We also note related work [ref1] that explores the fusion of Mixtral, InternLM2, and OpenChat, reporting consistent performance improvements under the knowledge fusion paradigm. We thus conduct a preliminary experiment on a MiniPile subset to extend our framework to fuse additional LLMs. Specifically, in this additional experiment, our approach is trained on a subset of MiniPile (100K training examples in total) created by randomly sampling 10% of the original dataset.
>
> The results presented in the table below demonstrate that our approach exhibits robustness and effectively integrates knowledge from diverse source models. Due to the constrained computational resources and time limit, these experiments are ongoing. We will provide updated results once they have been completed. We are excited to explore the training on full-size MiniPile further.
>
> |      | OpenLLaMA | MPT   | Llama-2 | Mistral 7B | SOLAR 10.7B | OpenChat 3.5 7B | Qwen2.5-7B | PTA-LLM (ours) | PTA-LLM + Mistral | PTA-LLM + Qwen2.5 | PTA-LLM + SOLAR | PTA-LLM + OpenChat 3.5  |
> | ---- | --------- | ----- | ------- | ---------- | ----------- | --------------- | ---------- | -------------- | ------------------------ | ------------------------ | ---------------------- | ----------------------------- |
> | MMLU | 42.11     | 27.84 | 46.94   | 59.15      | 63.05       | 61.58           | 71.80      | 47.42          | 48.69                    | 51.28                    | WIP                    | WIP                           |
>
> [ref1] FuseChat: Knowledge Fusion of Chat Models
>
> **Q2.1: Regarding the improvement over FuseLLM.**
>
> Thank you for the great observation. While our average absolute gain is 0.49% compared to the FuseLLM, the relative gain of 1.72% is nontrivial. Moreover, our significance test results show that the improvements are statistically significant with p-value < 0.005. We also reported the standard deviation over three runs in the table below.
>
> |                    | MMLU | BBH  | MultiPL-E |
> | ------------------ | ---- | ---- | --------- |
> | standard deviation | 0.05 | 0.04 | 0.05      |
>
> Our findings indicate that the performance remains stable once the hyperparameters and training devices are fixed. Other benchmarks show similar trends on standard deviation. We have included these results in the revised paper.
>
> **Q2.2: Regarding the evaluation of GSM.**
>
> Thank you for the insightful suggestion. Following your suggestions, we reevaluated our method using an 8-shot setting under a widely adopted evaluation framework, lm-evaluation-harness. The gsm8k-cot evaluation results for Llama-2 7B (i.e., 14.18%) align closely with those reported in the original paper [ref1] (i.e., 14.60%).
>
> |       | OpenLLaMA | MPT  | Llama-2 | Llama-2 CLM | FuseLLM | PTA-LLM        |
> | ----- | --------- | ---- | ------- | ----------- | ------- | -------------- |
> | GSM8K | 7.81      | 9.17 | 14.18   | 14.33       | 14.56   | 14.71 (+1.03%) |
>
> From the results, we observe a consistent performance improvement when utilizing probabilistic token alignment. We have revised the reported results to ensure a fairer and more general comparison.
>
> [ref1] Llama 2: Open Foundation and Fine-Tuned Chat Models
>
> (Due to character limitations, we will continue to answer in another comment.)

---

> ### Author Response · Authors · 2024-11-21
>
> (Continue from where rebuttal ends.)
>
> **Q3: Regarding the study of stability.**
>
> Thank you for the great question. In addition to the performance comparisons detailed in Table 2, we provide supplementary case studies in Appendix D to offer deeper insights into the effectiveness of our proposed method. For instance, consider the *Tracking Shuffled Objects* task with seven objects, a particularly challenging scenario requiring accurate tracking of the corresponding dancers among seven individuals as they switch partners many times.
>
> In this context, FuseLLM fails to track the objective during the fourth partner switch, whereas PTA-LLM successfully tracks the corresponding dancers throughout. This superior performance is likely attributable to PTA-LLM's probabilistic token alignment mechanism, which effectively transforms logits into the correct objective rather than merely replicating the original logits from FuseLLM. We have provided an expanded discussion of this analysis in the revised paper.
>
> **Q4: Regarding the ablation experiments.**
>
> Following your suggestion, we provide additional settings for each hyperparameter to reveal the impact trends of each factor. As observed from the table, within a reasonable range, our method is robust. We have supplemented these experiments with further discussion in the revision. Thank you again for your suggestion.
>
> | Optimal Transport Convergence Threshold | BBH   | ME    | MMLU  |
> | ----------------------------- | ----- | ----- | ----- |
> | 1e-3                          | 39.44 | 15.10  | 48.23 |
> | 1e-4                          | 40.54 | 15.88 | 48.99 |
> | 5e-5                          | 40.91 | 15.85 | 49.32 |
> | 1e-5                          | 41.08 | 15.82 | 49.38 |
> | 1e-6                          | 41.04 | 15.78 | 49.35 |
> | 1e-7                          | 41.05 | 15.80  | 49.33 |
>
> | Token Alignment Window Size | BBH   | ME    | MMLU  |
> | --------------------------- | ----- | ----- | ----- |
> | 10                          | 41.08 | 15.88 | 48.99 |
> | 7                           | 40.99 | 15.73 | 49.00    |
> | 5                           | 40.68 | 15.61 | 49.38 |
> | 3                           | 39.64 | 15.08 | 47.11 |
>
> | Combination weight | BBH   | ME    | MMLU  |
> | ------------------ | ----- | ----- | ----- |
> | 0.9                | 40.39 | 15.72 | 48.93 |
> | 0.85               | 41.00    | 15.91 | 49.09 |
> | 0.8                | 41.08 | 15.88 | 49.38 |
> | 0.75               | 39.78 | 15.65 | 47.29 |
> | 0.7                | 38.11 | 14.27 | 46.08 |
> | 0.6                | 37.20  | 14.09 | 45.96 |
>
> **Q5: Regarding the typos.**
>
> Thank you for pointing them out. We have revised accordingly and reformulated all the equations to ensure clarity and precision.
>
> We appreciate your thoughtful comments. We hope our response addresses your concerns. Please let us know if there are any additional questions, and we will be happy to discuss further.

---

> > ### Comment · Reviewer_qifk · 2024-11-24
> > **Official Comment by Reviewer qifk**
> >
> > Thank you for providing further explanations and additional experimental results.
> >
> > Regarding your response to Q1, the new experimental results show that the performance of the PTA-LLM combined with Mistral and Qwen-2.5 on MMLU does not surpass the original performance of Mistral (48.69 < 59.15) or Qwen-2.5 (51.28 < 71.80). This suggests that the proposed token alignment method does not effectively enhance performance when applied to heterogeneous models. Additionally, evaluations on heterogeneous models should include generative tasks, as MMLU predominantly focuses on multiple-choice questions. While I understand the challenges related to time and computational resources, I believe that such experiments are critical for demonstrating the effectiveness of the proposed method.
> >
> > Regarding your response to Q2, I remain unconvinced that an absolute improvement of less than 1% is sufficient to demonstrate the effectiveness of PTA-LLM.
> >
> > Regarding your response to Q3, I appreciate the explanation of stability through a case study, which has helped me better understand your method.
> >
> > Regarding your response to Q4, the new experimental results reveal no clear correlation between the three sets of hyper-parameters and the outcomes. This observation conflicts with the description provided in Section 4.5 of the paper, where a stronger relationship was implied. Moreover, under certain settings, the performance of PTA-LLM is even lower than that of FuseLLM, which further deepens my concerns about the effectiveness of PTA-LLM.
> >
> > I sincerely thank the authors for their detailed responses and efforts. However, after carefully considering the above points and the overall discussion, I have decided to keep
> > my current score for this submission. Thank you for your understanding.

---

> ### Author Response · Authors · 2024-11-24
> **Thank you for the prompt response**
>
> We sincerely appreciate the reviewer's engagement in the discussion and their valuable comments, which are crucial in improving the quality of our work. We’d like to take the opportunity to provide further clarifications regarding Q1 and Q4.
>
> **Regarding Q1**
>
> **First**, we completely agree that our experiment would benefit from incorporating models with greater architectural diversity. Our initial choice to include three models was based on the FuseLLM setting to ensure a fair comparison.
>
> **Second**, please note that the results reported above were obtained **using only 10% of the data** to quickly generate preliminary findings. The primary aim was to demonstrate that fusing additional models can indeed improve performance compared to PTA-LLM. For instance, PTA-LLM + Qwen2.5 improves MMLU performance from 47.42 to 51.28, achieving an absolute ~4% improvement with just 10% training data. We have since completed training on 100% of the data and present fair comparisons with Mistral and Qwen2.5 below. The results show that both PTA-LLM + Qwen2.5 and PTA-LLM + Mistral outperform Qwen2.5 and Mistral, respectively, demonstrating the effectiveness of our approach.
>
> |  100% Data    | OpenLLaMA | MPT   | Llama-2 | FuseLLM |  PTA-LLM (ours) | Mistral 7B | **PTA-LLM + Mistral** | Qwen2.5-7B | **PTA-LLM + Qwen2.5** |
> | ---- | --------- | ----- | ------- | ---------- | ----------- | --------------- | ---------- | -------------- | ------------- |
> | MMLU (multiple_choice) | 42.11     | 27.84 | 46.94   | 48.77      | 49.38       | 59.15          | **60.34**      | 71.80          | **72.66**                  |
> | Hellaswag (multiple_choice) | 74.52 | 76.35 | 75.99   | 78.23      | 79.74       | 80.39         | **81.52**      | 78.89          | **80.13**                  |
>
> **Third**, for generative tasks, we have already included results on MultiPL-E in the original paper. To provide further comparisons with the Mistral and Qwen models, we conducted additional experiments on HumanEval using MiniPile+Github for training. The results below, combined with our original findings on MultiPL-E, demonstrate that our method is effective for generative tasks as well.
>
> |  MiniPile + Github  | Mistral 7B | **PTA-LLM + Mistral** | Qwen2.5-7B | **PTA-LLM + Qwen2.5** |
> | ---- | --------- | ----- | ------- | ---------- |
> | HumanEval (generative task) | 30.50     | **32.64** | 57.90   | **59.71**   |
>
> **Regarding Q4**
>
> Thank you for these excellent questions. We’d like to clarify that the additional experiments are consistent with the findings presented in Section 4.5. Specifically:
>
> 1. **transport convergence threshold**: The findings on the optimal transport convergence threshold align with our motivation. Specifically, a lower threshold preference suggests that stricter constraints may generate a more coherent fusion, leading to greater performance gains.
>
> 2. **token alignment window size**: The findings of larger token alignment window sizes lead to better performance is consistent with previous results. A larger transport range allows for a more comprehensive understanding of the context, which in turn improves performance.
>
> 3. **combination weight**: Sorry for the confusion on this part. The paper mentions "higher performance when the weight is smaller" is referring to a comparison between 0.8 and 0.9. However, if the weight is reduced further (e.g., to 0.6), the model **overemphasizes** the fused matrix and pays less attention to the original CLM modeling. Consequently, we selected 0.8 for all experiments, as it consistently achieves the best performance. Thank you for highlighting this point; we have revised the paper accordingly.
>
> We greatly value each comment and suggestion from your review, and are hoping that our additional clarifications and experimental results address your concerns. We are eager to address any remaining issues during the discussion phase. Thank you very much.

---

> > ### Comment · Reviewer_qifk · 2024-11-25
> > **Official Comment by Reviewer qifk**
> >
> > Thank you for your supplementary information and detailed explanation. After reviewing your response, I have decided to raise my score to 5 points. I hope that the insights from this discussion can be incorporated into the next version of the paper to help readers better understand your work.

---

> > > ### Author Response · Authors · 2024-11-25
> > > **Thank you for engaging the discussion!**
> > >
> > > Thank you for your response and for raising the scores. We're glad our responses have addressed your concerns. We will make sure to incorporate all the discussion points into the revision as suggested.

---

### Official Review · Reviewer_HyYP · 2024-10-31

**Soundness:** 2
**Presentation:** 3
**Contribution:** 2
**Rating:** 6
**Confidence:** 2

**Summary:**

This paper introduces PTA-LLM, a probabilistic token alignment method designed to align the distributions of various LLMs with different vocabularies. Inspired by the optimal transport problem, PTA-LLM employs distribution-aware learning to enhance more coherent model fusion. Experimental results across multiple benchmarks confirm the effectiveness of PTA-LLM.

**Strengths:**

- The paper is easy to read and presents its ideas clearly.

- The proposed PTA-LLM is general, stable, and interpretable.

- The paper conducts a series of experiments to demonstrate the effectiveness of PAT.

**Weaknesses:**

- The improvement in effectiveness seems insufficiently significant. In Table 1, the average gap between CLM is only about 1 point. Given the complexity of Model Fusion compared to ordinary CLM, this improvement may not adequately highlight the advantages of PTA. For example, PAT requires obtaining the probability distributions of all teacher models, which means incurring multiple forward pass costs. What would be the effect if these additional costs were used to train more data with CLM? In addition, Multiple experiments are needed for significance testing to demonstrate the effectiveness of PAT.

**Questions:**

- Why does llamda2 perform only 0.66 on GSM, while I saw that the original llamda2 paper did not report such a low score?

- Why is direct training for next token prediction not ideal? In theory, all models fit the training corpus through next-token prediction. Why does distilling the distribution of different models on the corpus outperform the next-token-prediction training objective itself?

- Knowledge fusion also keeps the CLM loss for training, using a parameter called lambda to control weight. How does the value of lambda affect the performance? Whether lambda need to be fine-tuned elaborately for improvement？

---

> ### Author Response · Authors · 2024-11-21
>
> Dear reviewer HyYP,
>
> We sincerely appreciate your time and effort in reviewing our paper and providing valuable comments. We provide explanations to your questions point-by-point in the following.
>
> **Q1: Regarding the effectiveness.**
>
> Thank you for your insightful questions.
>
> **Regarding the performance and statistically significant.** While our average absolute gain is 1.1% compared to the CLM, the relative gain of 3.94% is nontrivial. In addition, our significance test results indicate that the improvements are statistically significant with p-value < 0.005. We further rerun the same hyperparameter settings three times and reported standard deviation error bars.
>
> |                    | MMLU | BBH  | MultiPL-E |
> | ------------------ | ---- | ---- | --------- |
> | standard deviation | 0.05 | 0.04 | 0.05      |
>
> Our findings indicate that the performance remains stable once the hyperparameters and training devices are fixed. Other benchmarks show similar trends on standard deviation. We have included these results in the revised paper.
>
> **Regarding using the additional costs to train more data.**
>
> This is an excellent question. To further demonstrate the effectiveness of our method, we conducted an additional experiment to compare two distinct settings following your suggestion.
>
> Setting A: Train on a subset of MiniPile (100K examples by randomly sampling 10% of the original dataset) using the same setting in the paper. Specifically, the total training costs, including obtaining the probability distributions of all teacher models (approximately 3.2 GPU hours) and end-to-end training (approximately 20 GPU hours), are 23.2 GPU hours.
>
> Setting B: Use the additional cost to train on more data. Specifically, we train Llama-2 CLM on a larger MiniPile subset (116K training examples, representing 11.6% of the original dataset) to match the total training costs of Setting A (23.2 GPU hours).
>
> The results, summarized in the table below, are consistent with our findings, demonstrating the benefits of the knowledge fusion paradigm under equivalent resource constraints. We sincerely appreciate your constructive suggestions.
>
> | Same Cost | Setting A (Our method) | Setting B (CLM with more data) |
> | -------------- | ---------------- | ------------------------------ |
> | MMLU           | 47.42            | 46.33                          |
>
> **Q2: The explanation of the EM score of GSM.**
>
> Sorry for the confusion. The low score reported is attributed to our initial evaluation being conducted under zero-shot settings without employing chain-of-thought prompting, which does not comprehensively reflect the true capabilities of the evaluated models.
>
> To address this issue, we reevaluated our method using an 8-shot setting under a widely adopted evaluation framework, lm-evaluation-harness. The gsm8k-cot evaluation results for Llama-2 7B (i.e., 14.18%) closely align with the result reported in the original paper [ref1] (i.e., 14.6%). From the table below, we observe consistent performance improvements when utilizing probabilistic token alignment.
>
> |       | OpenLLaMA | MPT  | Llama-2 | Llama-2 CLM | FuseLLM | PTA-LLM        |
> | ----- | --------- | ---- | ------- | ----------- | ------- | -------------- |
> | GSM8K | 7.81      | 9.17 | 14.18   | 14.33       | 14.56   | 14.71 (+1.03%) |
>
> We have revised the reported results to ensure a fairer and more general comparison. Thank you for your question.
>
> [ref1] Llama 2: Open Foundation and Fine-Tuned Chat Models
>
> (Due to character limitations, we will continue to answer in another comment.)

---

> > ### Comment · Reviewer_HyYP · 2024-11-25
> > **Response to Author**
> >
> > Thank you for your response. I am still confused about the GSM score. According to the llama2 report, llama2-7b has an average score of 14.6 on GSM and MATH. You reported a GSM score of 14.18. Since MATH is much harder than GSM, the GSM score should be significantly higher than 14.6.
> >
> > “MATH. We report the average of the GSM8K (8 shot) (Cobbe et al., 2021) and MATH (4 shot) (Hendrycks et al., 2021) benchmarks at top 1.” --llama2 report

---

> ### Author Response · Authors · 2024-11-21
>
> (Continue from where rebuttal ends.)
>
> **Q3: Why does knowledge fusion outperform the CLM objective?**
>
> Thank you for the insightful question. Two potential factors may explain why the knowledge fusion objective outperforms the traditional CLM approach: **First**, the CLM objective employs one-hot vectors as the golden labels, which fails to capture the nuanced information each token might convey. This approach provides the same penalty for completely incorrect predictions as for predictions that select an incorrect token but retain semantically relevant context. In other words, the CLM objective does not reward predictions that are "almost correct," which limits its capacity to encourage fine-grained improvements. **Second**, the fusion objective incorporates representations from diverse source models through distillation, enabling it to capitalize on the complementary strengths of each model. It provides more fine-grained context information for alignment. We have expanded on these points further in the revised version of the manuscript, offering a more comprehensive discussion.
>
> **Q4: Regarding the combination weight.**
>
> Thank you for your question. In addition to the ablation study presented in Section 4.6, we conducted supplementary experiments to further analyze the impact of the combination weight hyperparameter.
>
> | Combination weight | BBH   | ME    | MMLU  |
> | ------------------ | ----- | ----- | ----- |
> | 0.9                | 40.39 | 15.72 | 48.93 |
> | 0.85               | 41.00    | 15.91 | 49.09 |
> | 0.8                | 41.08 | 15.88 | 49.38 |
> | 0.75               | 39.78 | 15.65 | 47.29 |
> | 0.7                | 38.11 | 14.27 | 46.08 |
> | 0.6                | 37.20  | 14.09 | 45.96 |
>
> We observe consistent performance improvements when λ is set within the range of 0.8 to 0.9. Furthermore, we conclude that different benchmarks exhibit slight variations in hyperparameter preferences (i.e., BBH and MMLU tend to favor λ=0.8, while ME shows a preference for λ=0.85). Additionally, our method has a preference for lower λ values in comparison to FuseLLM (i.e., 0.9), suggesting a stronger emphasis on probabilistic token alignment in our method.
>
> We appreciate your thoughtful comments. We hope our response addresses your concerns. Please let us know if there are any additional questions, and we will be happy to discuss further.

---

> ### Author Response · Authors · 2024-11-25
> **Thank you for the prompt response**
>
> Thank you for the additional question. This is an excellent observation, and we’d like to provide some explanation here.
>
> First, in the LLaMA2 report [ref1], **Table 25 (page 51)** presents separate scores for GSM8k (14.6) and MATH (2.5) of LLaMa2-7B. The GSM8k score of 14.6 also aligns with the results listed on their official website (https://www.llama.com/llama2/). To confirm this further, we re-ran LLaMA2-7B on GSM8k using the officially released model [ref2] and obtained a consistent result of 14.18.
>
> Based on this, we suspect there may be a small inconsistent issue with the results reported in Table 3 of the LLaMA2 report. Specifically, the **Math** results in Table 3 indeed represent the average of the GSM8k and MATH scores from their Table 25, **except** for the LLaMA2-7B and LLaMA2-14B models.
>
> We greatly appreciate your feedback and hope these clarifications address your concerns. Please let us know if there are any other issues or questions you'd like us to address during the discussion phase. Thank you again for your valuable input!
>
> [ref1] Llama 2: Open Foundation and Fine-Tuned Chat Models (https://arxiv.org/pdf/2307.09288)
>
> [ref2] https://huggingface.co/meta-llama/Llama-2-7b-hf

---

> > ### Comment · Reviewer_HyYP · 2024-11-26
> > **Response to Author**
> >
> > Thank you for resolving my confusion, I have raised the score to 6.

---

> > > ### Author Response · Authors · 2024-11-26
> > > **Thank you for engaging the discussion!**
> > >
> > > We are delighted to hear that we have successfully addressed all your concerns and received your approval of the work.

---

### Official Review · Reviewer_biko · 2024-11-03

**Soundness:** 3
**Presentation:** 3
**Contribution:** 3
**Rating:** 6
**Confidence:** 3

**Summary:**

This paper proposes PTA-LLM, a knowledge fusion-based approach for language model fusion. Compared with FuseLLM, a pioneer knowledge fusion work, PTA-LLM changes the hard mapping of source and target vocabulary into soft mapping and uses an optimal transport method to calculate the optimal token alignment. Overall, the paper offers a sound and well-motivated solution to the core problem, token alignment, in knowledge fusion over heterogeneous source LLMs.

**Strengths:**

- Token alignment is the core issue of heterogeneous knowledge fusion. The paper offers a sound and well-motivated solution to this problem.
- The overall experimental results look good, though the improvements are not significant.

**Weaknesses:**

- In practice, the Sinkhorn algorithm seems to be time-consuming. The limitations section should be better included in the main pages for a fair demonstration. I also have some concerns about the end-to-end training time on corpus other than MiniPile.
- What does the EM score for GSM stand for? Is it 1.90%? The score seems extremely low and does not align with other reported results.

**Questions:**

None

---

> ### Author Response · Authors · 2024-11-21
>
> Dear reviewer biko,
>
> We sincerely appreciate your time and effort in reviewing our paper and providing valuable comments. We provide explanations to your questions point-by-point in the following.
>
> **Q1: Regarding the limitations section.**
>
> Thank you for the insightful suggestion. We have incorporated this limitation into the main manuscript accordingly.
>
> **Q2: Regarding the end-to-end training time.**
>
> Since the end-to-end training time primarily depends on the dataset size, we conducted an additional study using subsets of MiniPile (which contains 1M training samples). Specifically, we randomly sampled the original MiniPile to create subsets for training.
>
> |               | MiniPile 1M (100%) | MiniPile 500K (50%) | MiniPile 100K (10%) | MiniPile 10K (1%) |
> | ------------- | ------------------ | ------------------- | ------------------- | ----------------- |
> | Training Time | 26 hours           | 12.9 hours          | 2.4 hours           | 0.3 hours         |
>
> The results demonstrate a linear relationship between dataset size and training time. Experiments were conducted on 8 NVIDIA A100-80GB GPUs.
>
> To provide further information, we have conducted training on various size datasets and reported their estimated time computed by the framework on 8 NVIDIA A100-80GB GPUs. The same linear relationship can also be observed from the table.
>
> | Pile           | Openwebtext-en | Wikipedia-en | MiniPile |
> | -------------- | ------------------ | ---------------- | -------- |
> | Training Size  | 8M                 | 6M               | 1M       |
> | Estimated Time | 220 hours          | 168 hours        | 26 hours |
>
> [ref1] https://huggingface.co/datasets/vietgpt/openwebtext_en
>
> [ref2] https://huggingface.co/datasets/wikimedia/wikipedia
>
> **Q3: The explanation of the EM score of GSM.**
>
> Thank you for the insightful observation. We’d like to provide some clarification here. The EM (Exact Match) score for GSM was intended to measure the accuracy of predicted mathematical results, requiring an exact match with the ground truth answer. The low score reported is attributed to our initial evaluation being conducted under the **zero-shot** setting without employing chain-of-thought prompting.
>
> To further address your concern, we have reevaluated our method using an 8-shot setting under a widely adopted evaluation framework, lm-evaluation-harness. As shown in the results below, the gsm8k-cot evaluation results for Llama-2 7B (i.e., 14.18%) closely align with the result reported in the original paper [ref1] (i.e., 14.6%). From the table below, we observe consistent performance improvements when utilizing probabilistic token alignment. We have revised the reported results in Sec. 4.2 for completeness.
>
> |       | OpenLLaMA | MPT  | Llama-2 | Llama-2 CLM | FuseLLM | PTA-LLM        |
> | ----- | --------- | ---- | ------- | ----------- | ------- | -------------- |
> | GSM8K | 7.81      | 9.17 | 14.18   | 14.33       | 14.56   | 14.71 (+1.03%) |
>
> [ref1] Llama 2: Open Foundation and Fine-Tuned Chat Models
>
> We appreciate your thoughtful comments. We hope our response addresses your concerns. Please let us know if there are any additional questions, and we will be happy to discuss further.

---

> > ### Comment · Reviewer_biko · 2024-11-25
> > **Thank for your response**
> >
> > Thank you for your timely response. I have no further questions and I would like to keep my score.

---

### Official Review · Reviewer_vEFt · 2024-11-04

**Soundness:** 3
**Presentation:** 3
**Contribution:** 3
**Rating:** 8
**Confidence:** 3

**Summary:**

This paper proposes a new method for aligning tokens when fusing models with different vocabularies. The method uses optimal transport to transfer the information from a source model’s logits to the target model’s logits for a given text. Dynamic token pairing allows a single token from the shorter token sequence to be mapped to multiple tokens in the other’s sequence. The transport plan minimizes the cost of transferring mass between the two logit spaces according using edit distance of tokens to determine transport cost. The most probable alignment is selected, and the token logits of the target model are adjusted toward their aligned source token logits.

Positive evidence for the value of this method is provided through experiments on a variety of standard LLM evaluation datasets compared to reasonable baselines and related work. Analysis shows that the proposed method provides more stability than other fusion methods. A qualitative analysis of the proposed method shows a more compact and consistent representation compared to a competitive fusion method. Ablations provide information about the robustness and effect of hyperparameters.

**Strengths:**

This paper provides an interesting solution to the problem of mixing information from language models with different vocabularies. The performance gains from the proposed method are consistent versus the competing methods. Optimal transport is very fashionable right now, and the ICLR community may appreciate seeing it applied here.

**Weaknesses:**

The paper’s writing is at times grandiose. The introduction begins with a question about how to build better models without resorting to scaling, and then insists we reframe this question as one about model fusion. Some hedging here would help i.e. “one method for addressing question 1 is knowledge fusion”.

Sometimes, the paper is unclear. For instance: while optimal transport is general and can be applied to moving mass between a variety of spaces, it is most often applied in ML to moving probability mass. The current paper uses the standard ML terminology of “transferring probability” between distributions (e.g. line 272), but seems to be moving mass between logit spaces. Since logits are not probability distributions, the use of terms like “token distribution” and “probability mass” are a bit misleading. I’m not certain that moving mass in logit-space is the same as moving mass between probability spaces since the normalization terms in the source and target space are different. More careful writing could address this problem.

**Questions:**

The paper argues the proposed method improves on earlier methods which rely on “surface level token correspondences” (line 153), but doesn’t the use of edit distance as a cost function for the transport plan also rely on surface level correspondences? Is this edit distance computed at the token level?

---

> ### Author Response · Authors · 2024-11-21
>
> Dear reviewer vEFt,
>
> We sincerely appreciate your time and effort in reviewing our paper and providing valuable comments. We provide explanations to your questions point-by-point in the following.
>
> **Q1: Regarding the writing.**
>
> Thank you for your suggestion. We have revised our writing to adopt a more accurate and modest tone.
>
> **Q2: Regarding use of terms like ‘token distribution’ and ‘probability mass’ are a bit misleading.**
>
> Sorry for the confusion. We completely agree with the reviewer that conducting transport in logit space differs fundamentally from transporting mass in probability space due to the distinct normalization terms associated with the source and target spaces. In this work, we perform optimal transport on logits after applying the softmax function to reduce the impact of extreme values (e.g., extreme large, small, or negative values) that could otherwise distort the transport cost. Therefore, the term "transport probability" more accurately describes our method. We have revised accordingly to make it more clear and sincerely appreciate your thoughtful comments.
>
> **Q3: The explanation of "surface-level token correspondences”.**
>
> Yes, the edit distance is computed at the token level in our method as well. However, besides using edit distance as one metric, our method advances the distance measurement by incorporating the corresponding logit values into the individual cost within the transport framework. Optimization is thus performed at both the "surface-level" and "logit-level".
>
> Specifically, our cost function incorporates **both** the **edit distance cost** and the **corresponding logit values (i.e., $c_{xy}p_{xy}$),** with the overall transport cost being the summation over all alignments: $\sum \sum c_{xy}p_{xy}$. We have revised the content to make it more clear.
>
> We appreciate your thoughtful comments. We hope our response addresses your concerns. Please let us know if there are any additional questions, and we will be happy to discuss further.

---

> > ### Comment · Reviewer_vEFt · 2024-11-24
> >
> > Thanks for your responses.

---

### Author Response · Authors · 2024-11-21
**Summary of Revision**

To All Reviewers:

Thank you for your thorough review and insightful comments. We have revised our paper according to the suggestions. The major changes are summarized as follows:

- As suggested by Reviewer vEFt, we have revised our writing in Sections 1 and 2 to adopt a more accurate and modest tone. Additionally, we added more discussion about the terminology of "transport probability" in Appendix A.
- Following the comments from Reviewers biko, HyYP, and qifk, we revised the reported results in Section 4.2 to ensure a fairer and more general comparison. Moreover, we supplemented experiments with more hyperparameter settings in Appendix E and expanded the discussion in Appendix F to further clarify why knowledge fusion outperforms the CLM objective.
- As suggested by Reviewer biko, we have incorporated the limitation into Section 5 to ensure a fair demonstration.
- Based on the suggestions by Reviewer qifk, we have corrected typos in Section 3.2 and provided an expanded discussion of case studies in Appendix D. Furthermore, we reported the p-value and standard deviation in Appendix C to demonstrate the statistical significance of our results.
- We updated Figures 1-3 to enhance their interpretability and alignment with the revised text. Additionally, all equations (1-5) were reformulated for improved clarity and precision.

All modifications have been marked in ${\color{blue} blue}$ in our revised submission.

Sincerely yours,\
Authors

---

### Author Response · Authors · 2024-11-23
**Looking forward to the discussion**

Dear Reviewers,

We sincerely appreciate the time and effort you've devoted to reviewing our work. We understand that your schedule may be quite busy, and we are truly grateful for your valuable feedback. As the Author-Reviewer discussion phase is ending soon, we would greatly value the opportunity to engage in further discussion with you. Our aim is to gain insights into whether our responses effectively address your concerns and to ascertain if there are any additional questions or points you would like to discuss.

We look forward to the opportunity for further discussion with you. Thank you for your thoughtful consideration.

Best regards,\
Authors

---

### Author Response · Authors · 2024-12-03
**Summary of Author-Reviewer Discussion**

Dear Area Chair and Reviewers,

We would like to express our sincere gratitude for your efforts in facilitating the discussion regarding our paper. As the discussion is coming to an end, we would like to provide a brief summary of the key points that have been discussed:

- We have clarified the terms of "token distribution" and "probability mass" in the revised paper to enhance their clarity, as suggested by **Reviewer vEFt**.

- In response to **Reviewer biko**'s suggestions, we have included an analysis of training time with varying data sizes. Additionally, we have elaborated on the evaluation of the EM score on GSM and provided supplementary experimental results under 8-shot settings.

- As recommended by **Reviewer HyYP**, we have included significance test results and conducted further experiments on cost-effective training under identical resource constraints. These results further validate the superior performance of our approach.

- Following **Reviewer qifk**'s suggestions, we have conducted experiments integrating additional heterogeneous LLMs, showcasing the robustness of our method in fusing diverse source models. We have also added supplementary case studies to illustrate the stability and effectiveness of our approach, along with detailed ablation studies analyzing the impact of each hyperparameter.

In summary, we are grateful to all reviewers for their constructive feedback and acknowledgment of our responses. We particularly appreciate that **Reviewers HyYP and qifk** have increased their scores and that **Reviewers vEFt and biko** have maintained their positive evaluations.

Finally, we deeply value the reviewers' insightful comments, which have helped us refine our work. We believe that our revisions significantly enhance the clarity and impact of our contributions. We hope our work provides meaningful insights and advances the development of the knowledge fusion domain.

Sincerely,\
Authors

---

### Meta-Review · Area_Chair_knPU · 2024-12-21

**Metareview:**

This paper tackles a technical challenge in knowledge fusion for LLMs: how do we align the outputs of different LLMs during the fusion step? When tokenization differs, the sequences produced by two LLMs may be different in length and use different vocabulary sizes / different tokens at each step.  The proposed approach uses a dynamic programming solution to find a potentially many-to-many monotonic alignment between the sequences. Once two tokens are aligned, an optimal transport problem is defined to yield a transport plan to align the distributions themselves.

The paper tackles an interesting (if somewhat niche) question and presents a novel and effective algorithm. The paper is generally well-written. The results show consistent improvement over FuseLLM.

One main weakness of this paper is the narrowness of the setting: this kind of fusion across LLMs with different tokenizers is a fairly particular setting.  Another major question is whether the accuracy improvements in Table 1 merit the complexity of the approach (HyYP, qifk). I am not convinced by relative gains. On most of the given tasks, even if there is an improvement over FuseLLM, the gain is not large enough to be practically significant: it may not transfer to noticeable performance improvements for LLMs in practice.

Runtime information is buried in the appendix and is left in theoretical terms except for a vague reference to 13.75% aligning delay. As biko points out, this should be included in the main body of the paper with more discussion and analysis, particularly as the accuracy improvements are small.

**Additional Comments On Reviewer Discussion:**

The biggest issues are the accuracy improvement (HyYP, qifk), which are addressed somewhat, but not satisfyingly to me: the gains *are* still small in all settings, regardless of their statistical significance or relative gains.

That said, extension to other models is shown in the response to qifk. Although the gains in the full data are small, they are nevertheless present, showing positive signal about the method.

Finally, the question of runtime is important and more discussion should be included in the paper, including the results from the discussion with biko.

There are also some questions raised about the GSM performance, which have been resolved.

---

### Decision · Program_Chairs · 2025-01-22

Reject